# Sharper Guarantees for Learning Neural Network Classifiers with Gradient Methods

**Hossein Taheri**
Department of Computer Science and Engineering,
University of California, San Diego.
htaheri@ucsd.edu

**Christos Thrampoulidis,**
Department of Electrical and Computer Engineering,
University of British Columbia.
cthrampo@ece.ubc.ca

**Arya Mazumdar**
Department of Computer Science and Engineering,
University of California, San Diego.
arya@ucsd.edu

## Abstract

In this paper, we study the data-dependent convergence and generalization behavior of gradient methods for neural networks with smooth activation. Our first result is a novel bound on the excess risk of deep networks trained by the logistic loss, via an alogirthmic stability analysis. Compared to previous works, our results improve upon the shortcomings of the well-established Rademacher complexity-based bounds. Importantly, the bounds we derive in this paper are tighter, hold even for neural networks of small width, do not scale unfavorably with width, are algorithm-dependent, and consequently capture the role of initialization on the sample complexity of gradient descent for deep nets. Specialized to noiseless data separable with margin $\gamma$ by neural tangent kernel (NTK) features of a network of width $\Omega(\text{poly}(\log(n)))$, we show the test-error rate to be $e^{O(L)}/\gamma^2 n$, where $n$ is the training set size and $L$ denotes the number of hidden layers. This is an improvement in the test loss bound compared to previous works while maintaining the poly-logarithmic width conditions. We further investigate excess risk bounds for deep nets trained with noisy data, establishing that under a polynomial condition on the network width, gradient descent can achieve the optimal excess risk. Finally, we show that a large step-size significantly improves upon the NTK regime's results in classifying the XOR distribution. In particular, we show for a one-hidden layer neural network of constant width $m$ with quadratic activation and standard Gaussian initialization that SGD with linear sample complexity and with a large step-size $\eta = m$ reaches the perfect test accuracy after only $\lceil \log(d) \rceil$ iterations, where $d$ is the data dimension.

## 1 Introduction

### 1.1 Overview

Neural networks, with their vast capacity for capturing intricate patterns in data, have triggered a paradigm shift in machine learning. Despite the power of these networks in modeling complex relationships, the interplay between their optimization and generalization behaviors (that is the gap between training and test errors) continues to be a compelling area of research. In practice, training neural networks using gradient-based optimization methods often leads to interpolation. That is, deep networks can meticulously fit the training data, driving empirical loss to near-zero and training error to perfect classification. However, these networks also demonstrate the capability to generalize well to unseen data. Various recent research endeavors have explored the training and test error guarantees of deep networks, with a focus on the Neural Tangent Kernel (NTK) regime Jacot et al. (2018); Du et al. (2019). One prominent feature of such works is that during gradient descent iterates the network's weights are constrained to move at most a constant distance with respect to overparameterization i.e.,

| | Activation | Width | Train Loss | Test Loss |
|---|---|---|---|---|
| Chen et al. (2021); Bartlett et al. (2017) | ReLU | $\Omega(\text{poly}(\frac{\log(n)}{\gamma}))$ | $\widetilde{O}(\frac{1}{\gamma^2 T})$ | $\widetilde{O}(\frac{e^{O(L)}}{\gamma^2}\sqrt{\frac{m}{n}} + \frac{1}{\gamma^2 T})$ |
| Thm. 2.1, Cor. 1 | Smooth | $\Omega(\text{poly}(\frac{\log(n)}{\gamma}))$ | $\widetilde{O}(\frac{1}{\gamma^2 T})$ | $\widetilde{O}(\frac{e^{O(L)}}{\gamma^2 n} + \frac{1}{\gamma^2 T})$ |

Table 1: Comparison of our results on learning deep nets with GD under NTK separability condition to related prior results. Here $m$ : network width, $L$ : network depth, $\gamma$ : NTK-margin, $n$ : number of samples and $T$ : number of iterations.

| | Activation | Width | Iteration | Sample |
|---|---|---|---|---|
| Telgarsky (2023); Chen et al. (2021); Taheri & Thrampoulidis (2024); Cor.1 | all | $\Omega(\text{poly}(d))$ | $d^2$ | $\widetilde{O}(d^2)$ |
| Glasgow (2024) | ReLU | $\Omega(\text{poly}(\log(d)))$ | $\text{poly}(\log(d))$ | $\widetilde{O}(d)$ |
| Thm. 2.4 | Quadratic | $\Omega(1)$ | $\log(d)$ | $\widetilde{O}(d)$ |

Table 2: Comparison of our findings on learning the $d$-dimensional XOR distribution with SGD to relevant prior results.

$\|w^\star - w_0\| = O_m(1)$, where $w^\star \in \mathbb{R}^p$ denotes the vector of target weights, $w_0$ is the initial weight vector, and $m$ is the network width Chen et al. (2021); Ji & Telgarsky (2020); Telgarsky (2023).

Yet, even for the relatively simple setting of learning deep nets in the kernel regime, the existing generalization bounds are still suboptimal. Moreover, the boundaries of the kernel regime are still largely unknown and an active area of research Liu et al. (2022); Banerjee et al. (2022); Telgarsky (2023). While the kernel regime can partially demonstrate the behavior of neural networks, the resulting guarantees often require large width, small step-size or large iteration and sample complexities. There is increasing evidence in recent years that for certain class of data distributions neural networks can overcome these limitations by using a large step-size which allows the network's parameters to move a long distance from initialization, often leading to better sample and computational complexities Damian et al. (2022); Ba et al. (2022).

In this work, we study the generalization and convergence behavior of gradient-based algorithms in neural nets with smooth activation functions for a wide class of data distributions. Our first result characterizes the test and train loss rates for classification problems under the condition that deviation from initialization is bounded depending on the network's width. In particular, for $L$-hidden layer networks our results hold under $\|w^\star - w_0\| \lesssim m^{O(1/L)}$, allowing the network's weights to move from initialization a distance increasing with $m$. This shows that the kernel regime continues to hold for a wider range of setups than previous results for which the deviations are restricted to be constant in $m$. The key reason for this improvement is exploiting the objective's Hessian structure in the gradient-descent path. More importantly, using the Hessian information enables us to develop, for the first time, *algorithm-dependent* generalization bounds of deep neural networks. As will be discussed throughout the paper, the bounds we derive via algorithmic-stability are tighter than previous relevant bounds in the literature. We specialize these results to a well-known NTK separability condition tailored for noiseless data and show that our results substantially improve the prior results on the test error performance while still allowing the width to be small, specifically poly-logarithmic on sample size (Table 1). A more detailed comparison is deferred to Section 1.2. We also consider the case of noisy data distributions and show that deep models are consistent, i.e., they can achieve the optimal test loss in the presence of noise as the sample size grows.

While these results improve upon the existing bounds within the NTK regime, we show in Section 2.2 that using a large step-size can further improve both the computational and sample complexities. In particular, we show for the stylized setup of data following the XOR distribution, a two-layer neural network with quadratic activation reaches zero test error after only $\log(d)$ steps of SGD with an aggressive step-size $\eta = m$. A comparison of our findings with the guarantees of the kernel regime with both GD Taheri & Thrampoulidis (2024) and one-pass SGD Telgarsky (2023); Chen et al. (2021) and the most relevant work in the feature learning regime Glasgow (2024) is summarized in Table 2.

Below is a summary of our contributions.

- In Theorem 2.1, we develop sufficient conditions for the global convergence of gradient descent in deep and smooth networks and show that if $m = \Omega(\|w^\star - w_0\|^{6L+4})$, the training loss is bounded by $O(\frac{\|w^\star - w_0\|^2}{\eta T})$, where $\eta$ is the step-size and $w^\star$ can be any choice of network weights that achieves small training loss. Under similar conditions on $m$, we show the generalization error is bounded by $O(\frac{\|w^\star - w_0\|^2 G_0^2}{n})$ where $G_0$ is the Lipschitz parameter of the network at initialization.
- In Corollary 1, we interpret these results by specializing them to a commonly-used margin-based NTK separability condition. The results of the corollary and comparison to previous works in literature are summarized in Table 1. A promising feature of our approach is that the test loss bound does not have an unfavorable dependence on the width while still maintaining minimum poly-logarithmic width conditions, which is new in the context of deep learning. To the best of our knowledge this is the tightest test error bound for deep nets trained by GD in the NTK regime.
- We consider the more general case of noisy data with non-vanishing optimal test loss in Theorem 2.3 and show that under a polynomial growth condition on network width, GD achieves a convergence rate of $1/\sqrt{n}$ to the optimal loss after $T = \sqrt{n}$ iterations.
- In Section 2.2, we consider the $d$-dimensional XOR distribution and show that a one-hidden layer network of constant width after exactly $\log(d)$ iterations of SGD with step-size $\eta = m$ achieves perfect test accuracy with $n = \widetilde{O}(d)$ samples, considerably improving kernel regime's limitations.

## 1.2 PRIOR WORKS

**Generalization of deep nets.** Among prior works on the generalization capabilities of deep networks, the only initialization dependent bounds were provided in Bartlett et al. (2017) obtaining bounds of order $O(\frac{\mathcal{R}}{n})$ where the Rademacher complexity is derived as $\mathcal{R} := (\prod_{i=1}^{L} \|W_i\|_2)(\sum_{i=1}^{L} \frac{\|W_i^\top - M_i^\top\|_{2,1}^{2/3}}{\|W_i\|_2^{2/3}})^{3/2}$. Here $\|W_i\|_2$ is the spectral norm of the weight matrix of layer $i$ (typically a constant) and $M_i$s are any data-independent matrices. Thus one can choose $M_i = W_{i,0}$, i.e., the initialization weight matrix. In fact, the above bound resembles the bound that we obtained via an optimization-dependent stability analysis. However, note that $\mathcal{R}$ depends on the distance traversed by weights through the $\ell_{2,1}$ norm which is always larger than the Frobenius norm, and in the worst case, the gap can be significantly large depending on the width. To see this, note that for a matrix $V \in \mathbb{R}^{m \times m}$ it holds that $\|V\|_{2,1} \leq \sqrt{m}\|V\|_F$. We note that "initialization-independent" bounds (e.g., Neyshabur et al. (2018); Golowich et al. (2018)) that are usually proportional to $\|w_t\|$ (rather than $\|w_t - w_0\|$) are strictly looser than the bound we obtain. This is primarily due to the fact that $\|w_t - w_0\|$ can be much smaller than $\|w_t\|$ and in fact as our experiments show $\|w_t - w_0\|$ is of constant order and can even decrease with width. Whereas, $\|w_t\|$ (or $\|w_t\|/\sqrt{m}$ due to the normalization in our setup) grows by increasing $m$, making the initialization-independent bounds potentially grow with width at the rate $O(\sqrt{m})$, despite lacking an explicit dependence on $m$. Hence, for wide networks, prior generalization bounds of deep neural nets based on Rademacher complexity are larger than the bound we derive in Theorem 2.1.

**Test rates under the NTK separability condition.** Other works that provide generalization bounds and optimization guarantees for neural nets include Cao & Gu (2019); Nitanda et al. (2019); Ji & Telgarsky (2020); Chen et al. (2021); Richards & Rabbat (2021); Taheri & Thrampoulidis (2024); Wang et al. (2023). In particular, Ji & Telgarsky (2020) derived the width condition $m = \Omega(\frac{\text{poly}(\log(n))}{\gamma^8})$ for achieving the $\frac{1}{\gamma^2 \sqrt{n}}$-test error rate in two-layer nets via a uniform-convergence argument Shalev-Shwartz & Ben-David (2014). This bound was extended to deep networks in Chen et al. (2021) with a generalization gap of order

$$\widetilde{O}\left(\frac{4^L}{\gamma^2}\sqrt{\frac{m}{n}} \wedge \left(\frac{L^{3/2}}{\gamma^2 \sqrt{n}} + \frac{L^{11/3}}{\gamma^2 m^{1/6}}\right)\right),$$

where $\wedge$ takes the minimum of two quantities. As discussed earlier, this bound is dependent on width since it is derived essentially by taking the minimum of two generalization bounds based on Rademacher complexity derived in Bartlett et al. (2017) and Cao & Gu (2019). Importantly, in the small width regime the bound simplifies into $\widetilde{O}(\frac{4^L}{\gamma^2}\sqrt{\frac{m}{n}})$, which has an undesirable dependence on

the width. In this paper, we improve the generalization gap to $\tilde{O}(\frac{e^{O(L)}}{\gamma^2 n})$ under the width condition $m = \Omega(\text{poly}(\log(n)/\gamma))$. To the best of our knowledge, these are the smallest generalization bound and width condition in literature up to now for learning deep neural nets. The key reason behind the improvement is leveraging the Hessian structure of the objective throughout the gradient descent iterates. The improved generalization guarantees result from the algorithmic dependency of our bounds and in fact the bounds can even be expressed such that they solely depend on the cumulative training loss (c.f. Eq. 5). As the training loss captures the role of initialization and is independent of width, the resulting generalization bounds share the same favorable properties.

**Feature learning and the XOR distribution.** Some recent works have pointed out the limitations of the kernel regime in understanding the full power of neural networks Abbe et al. (2022). In particular, as in the kernel regime, the networks weights are bounded not to move significantly from initialization, the learned features are not considerably different from those learned at initialization. On the other hand, it is hypothesized that neural networks can learn the true underlying features of the data distribution if the network weights are allowed (by large step-sizes or avoiding early-stopping) to move a large distance from initialization. This phenomenon was first proved for specific regression tasks where the labels essentially only depend on a small number of features, such as when $y = g(Ux)$ for $U \in \mathbb{R}^{k \times d}$ where $k \ll d$ Damian et al. (2022); Ba et al. (2022); Abbe et al. (2022); Cui et al. (2024), by one large SGD step leading to superior sample complexities compared to the kernel regime. For classification tasks, some focus has been on the XOR distribution(a.k.a. parities) Wei et al. (2019). Recent works have studied the problem of learning the $d$-dimensional XOR distribution using neural networks in both NTK and feature learning settings Barak et al. (2022); Telgarsky (2023); Taheri & Thrampoulidis (2024); Glasgow (2024). Specifically, it has been shown that under NTK with a sufficiently small step size, a polynomially wide network requires $d^2$ GD steps and $d^2$ sample size. Some studies have achieved linear sample complexity for learning XOR Bai & Lee (2019); Glasgow (2024); Telgarsky (2023); but these methods involve more computational effort compared to our results. The work most related to ours is Glasgow (2024), which demonstrated that with a particular Gaussian initialization, a ReLU network requires $\text{poly}(\log(d))$ large SGD steps and $\text{poly}(\log(d))$ neurons to learn XOR with linear sample complexity. In contrast, we show that by using a quadratic activation, learning this distribution requires only $\log(d)$ large steps with a constant-width network, while maintaining the same linear sample complexity.

## NOTATION

Probability and expectation with respect to the randomness in random variable $x$ are denoted by $\Pr_x(\cdot)$ and $\mathbb{E}_x[\cdot]$. We use $\mathbb{E}_x[w_t]$ to represent the weights after $t$ steps of gradient descent using the population's gradient. We use the standard complexity notation $\lesssim, o(\cdot), O(\cdot), \Theta(\cdot), \Omega(\cdot)$ and use $\tilde{o}(\cdot), \tilde{O}(\cdot), \tilde{\Theta}(\cdot), \tilde{\Omega}(\cdot)$ to hide polylogarithmic factors. We denote $a \wedge b := \min\{a, b\}$. The gradient and Hessian of the model $\Phi : \mathbb{R}^{p \times d} \to \mathbb{R}$ with respect to the first input (that is, weights) are denoted by $\nabla \Phi$ and $\nabla^2 \Phi$, respectively. The minimum eigenvalue of a symmetric matrix is denoted by $\lambda_{\min}(\cdot)$. We use $\|\cdot\|$ for the Euclidean norm of vectors and $\|\cdot\|_2$ for the spectral norm of matrices. We denote $[w_1, w_2] := \{w : w = \alpha w_1 + (1 - \alpha)w_2, \alpha \in [0, 1]\}$ the line between $w_1, w_2 \in \mathbb{R}^p$.

## 2 MAIN RESULTS

Throughout the paper, we consider the following unregularized objective for a neural network classifier parameterized with $w \in \mathbb{R}^p$,

$$\min_{w \in \mathbb{R}^p} \widehat{F}(w) := \frac{1}{n} \sum_{i=1}^n f\left(y_i \Phi(w, x_i)\right), \tag{1}$$

with data points satisfying $\|x\| \leq 1$, the binary labels $y_i \in \{\pm 1\}$ and $f(\cdot)$ is a loss function for classification tasks such as the logistic loss, $f(t) := \log(1 + e^{-t})$ and $\Phi(\cdot, x)$ is the network's output. We also define the test loss as $F(w) := \mathbb{E}_{x,y}[f(y\Phi(w, x))]$.

## 2.1 TRAIN AND TEST LOSS BOUNDS IN DEEP NETS

In our first theorem, we establish conditions for the width and target weights of the network that guarantees the training loss decays to zero if the network can interpolate the training set. We consider gradient descent update rule where at any iteration $t \leq T$: $w_{t+1} = w_t - \eta \nabla \widehat{F}(w_t)$. Before stating the theorem, we note that this result is valid under the standard descent-lemma condition for the step-size, as stated in Lemma 3 in the appendix. In particular, the descent lemma holds if the step-size satisfies the standard Eq. 34 in the appendix.

**Theorem 2.1** (Train & Test loss of deep nets). *Consider L-layer network where $\Phi(w, x) := \frac{1}{\sqrt{m}} W_{L+1}^\top (\frac{1}{\sqrt{m}} \sigma(W_L^\top \cdots \frac{1}{\sqrt{m}} \sigma(W_1^\top x) \cdots)$ and $\sigma$ is a 1-smooth and 1-Lipschitz activation function such that $\sigma(0) = 0$. Moreover, let $\beta_L$ be a constant that only depends on L, let all parameters of the network be initialized as i.i.d. standard Gaussian and assume the step-size satisfies the condition of the descent lemma. Fix T and assume the target weights vector $w^\star \in \mathbb{R}^p$ that obtain small training loss such that*

$$\rho^\star \geq \max \left\{ \sqrt{\eta T \widehat{F}(w^\star)}, \sqrt{\eta \widehat{F}(w_0)} \right\}. \tag{2}$$

*where $\rho^\star := \|w^\star - w_0\|$. Moreover, assume the width m is large enough such that it satisfies,*

$$m \geq 4\beta_L^2 (6\rho^\star)^{6L+4} \tag{3}$$

*Then, $\|w_t - w_0\| = O(\rho^\star)$ and the training loss satisfies with high probability over initialization,*

$$\widehat{F}(w_T) \leq \frac{4\rho^{\star 2}}{\eta T}. \tag{4}$$

*Moreover, assume for every n samples from the data distribution there exists $w^\star$ satisfying Eqs. 2-3. Then, the expected generalization gap satisfies with high probability over initialization,*

$$\mathbb{E}_{\mathcal{S}} \left[ F(w_T) - \widehat{F}(w_T) \right] \leq 2.2 \frac{\eta(G_0 + 1/4)^2}{n} \mathbb{E}_{\mathcal{S}} \left[ \sum_{t=0}^{T-1} \widehat{F}(w_t) \right], \tag{5}$$

*where the expectation is over the randomness in the training set denoted by $\mathcal{S}$ and $G_0$ is the Lipschitz parameter of network at initialization i.e., $\|\nabla \Phi(w_0, \cdot)\| \leq G_0$.*

In words, the main condition of the theorem is the existence of network weights denoted by $w^\star$ that achieves small training error (Eq. 2) and its distance from initialization is at most $O(m^{1/(6L+4)})$ as implied by Eq. 3. Under these conditions, the training loss is controlled solely by $\|w^\star - w_0\|$ and has no explicit dependence on the width or depth of the network. As it will be stated in Corollary 1, in the NTK regime with margin $\gamma$ it holds $\|w^\star - w_0\| = O(\log(n)/\gamma)$, leading to the width condition $m = \Omega(\text{poly}(\log(n)/\gamma))$. This is unlike previous results which either required polynomial width (such as (Liu et al., 2022; Cao & Gu, 2019)) or led to sub-optimal bounds (e.g., (Chen et al., 2021; Ji & Telgarsky, 2020)).

In general, the theorem is valid for any feasible minimizer $w^\star \in \mathbb{R}^p$. Thus, we can choose $w^\star$ with smallest value for $\|w^\star - w_0\|$ to optimize the bounds. With such choice of $w^\star$, the distance the weights obtained by GD travel is also minimized as $\forall t \in [T] : \|w_t - w_0\| = O(\|w^\star - w_0\|)$. Thus, gradient descent tends toward solutions which attain small loss and lie at minimum possible distance from initialization. This is in line with related prior observations in several other works such as (Du et al., 2019; Oymak & Soltanolkotabi, 2019).

The theorem also shows the sample complexity and iteration complexity of learning deep networks with gradient descent and demonstrates the role of initialization and weight's norms on the test error. Note that by replacing the time-averaged training loss guarantees (cf. Theorem B.1), the generalization gap simplifies into:

$$\mathbb{E}_{\mathcal{S}} \left[ F(w_T) - \widehat{F}(w_T) \right] \leq 9 \frac{\rho^{\star 2}(G_0 + 1/4)^2}{n}. \tag{6}$$

Hence, the test loss after $T = \Theta(n)$ iterations takes the form of

$$\mathbb{E}_{\mathcal{S}} \left[ F(w_T) \right] = O \left( \frac{\left\| w^\star - w_0 \right\|^2 G_0^2}{n} + \frac{\left\| w^\star - w_0 \right\|^2}{\eta n} \right), \tag{7}$$

where the first term is the generalization error and the second term is the training loss. Remarkably, Eq. 7 shows the tight correlation between the two terms, as the generalization gap is virtually the optimization error scaled by the squared Lipschitz constant $G_0^2$. This is indeed the consequence of Eq. 5 which bounds the generalization gap based on the cumulative optimization error.

At a high-level, the test loss essentially depends on two quantities: (i) the Euclidean distance between the target weights and the initialization and (ii) the Lipschitz parameter of network at initialization. Due to the algorithmic-dependent nature of our generalization bounds and unlike the prior bounds in literature, the bound in Eq. 7 captures the role of initialization on the test error: smaller deviations from initialization lead to smaller test error bounds and in particular the bound approaches zero as $\rho^\star$ goes to zero. As discussed earlier, gradient descent favors such solutions with small deviations from initialization. In addition to "distance from initialization", the test error bounds also depend on the squared Lipschitz parameter of the network. For standard Gaussian initialization, it can be shown (cf. Section A.2) that $G_0 \lesssim e^{O(L)}$ which introduces an exponential dependence on depth to the generalization bound. We remark that this dependence also appears in the corresponding bounds derived via uniform convergence and Rademacher complexity (e.g., (Bartlett et al., 2017; Golowich et al., 2018; Chen et al., 2021)) through the term $\prod_{i=1}^{L} \|W_i\|_2$.

An interesting feature of our approach is the algorithmic dependent bound in Eq. 5. This bound is generally tighter than the bound in Eq. 6. With the descent lemma condition on the step-size (c.f. Lemma 3) it holds that $\eta < 1/(G_0^2 + 1/4)$ which simplifies the bound into:

$$\mathbb{E}_{\mathcal{S}}\left[F(w_T) - \widehat{F}(w_T)\right] \leq \frac{2.2}{n} \mathbb{E}_{\mathcal{S}}\left[\sum_{t=0}^{T-1} \widehat{F}(w_t)\right]. \tag{8}$$

Hence, we have a bound which only depends on the training performance and the number of training samples. In our experiments in Section 3, we compute this bound for real-world data and compare it with the empirical results for generalization and test loss.

*Remark* 2.2. Our analysis relies on the recent progress in characterising the spectral norm of the deep net's Hessian during GD updates (Liu et al., 2020; Banerjee et al., 2022). In particular, Liu et al. (2020) proved that with standard Gaussian initialization, the model's Hessian is bounded with high probability by $\|\nabla^2\Phi(w, x)\| = O(\frac{R^{3L}}{\sqrt{m}})$ if $\|w - w_0\| \leq R$. These guarantees can also be used to study the convergence rate of deep networks trained by quadratic loss as done by (Liu et al., 2020; 2022) but their approach leads to excessively large width conditions. In contrast, here we consider classification tasks with an improved poly-logarithmic width requirement and also study the generalization performance.

Recently, the algorithmic stability has been employed for two-layer neural nets in (Taheri & Thrampoulidis, 2024; Richards & Rabbat, 2021) which is an improved analysis of the stability-based approach typically used for convex objectives in (Bousquet & Elisseeff, 2002; Hardt et al., 2016; Lei & Ying, 2020). Here, we essentially extend the stability analysis to deep networks. Compared to the two-layer nets, here in every iterate of gradient descent, the Hessian's norm guarantees depend on the network's weights. In particular, the analysis has to take into account that both $\|w_t - w_0\|$ and $\|w_t - w^\star\|$ remain small during GD updates. This is necessary in order to ensure the Hessian's norm guarantees and the approximate quasi convexity property hold during all GD iterates. We do this by an induction based argument which bounds these terms based on the fixed quantity $\|w^\star - w_0\|$, which is guaranteed to be bounded based on width by assumption.

### 2.1.1 Specializing to the NTK-separablity condition

The results in the last section can be specifically applied to a class of data distributions that includes the XOR distribution which will be discussed more through the rest of the paper. Before stating our result in the corollary, we state the neural tangent kernel (NTK) separability assumption.

**Assumption 1** (NTK-separability (Nitanda et al., 2019)). Assume the tangent kernel of the model at initialization separates the data with margin $\gamma$ i.e., for a unit-norm vector $w \in \mathbb{R}^p$ it holds for all $i \in [n] : y_i \langle \nabla\Phi(w_0, x_i), w \rangle \geq \gamma$.

In words, the above assumption implies that the features learned by the gradient at initialization can be linearly separated by some weights $w$. This assumption is commonly used in deep learning literature for studying classification tasks (Chen et al., 2021; Ji & Telgarsky, 2020).

**Corollary 1** (NTK results). *Consider the same setup as Theorem 2.1 and let Assumption 1 hold. Define constant $B > 0$ that bounds the model's output at initialization i.e., $\forall i \in [n] : |\Phi(w_0, x_i)| < B$. Assume the width is large enough such that $m \geq \beta_L^2 \left(\frac{2B + \log(1/\epsilon)}{\gamma}\right)^{6L+4}$. Then there exists $w^\star \in \mathbb{R}^p$ such that $F(w^\star) \leq \epsilon$ and $w^\star$ lies at a bounded Euclidean distance from initialization such that $\|w^\star - w_0\| \leq \frac{1}{\gamma}(2B + \log(1/\epsilon))$.*

To interpret this result, we apply this result to Theorem 2.1 and fix $\epsilon = 1/T$. First note that the output of network at initialization is constant with high-probability over initialization (e.g., see (Liu et al., 2020, Lemma F.4), (Cao & Gu, 2019, Lemma 4.4)) which implies $B = \tilde{O}(1)$. Overall, with the stopping condition $T = \Theta(n)$ and recalling that $G_0 \leq e^{O(L)}$, Corollary 1 yields the expected test error rate of order

$$\widetilde{O}\left(\frac{e^{O(L)}}{\gamma^2 n} + \frac{1}{\gamma^2 T}\right),$$

under the condition that $m = \Omega(\text{poly}(\log(n)) \cdot \beta_L / \gamma^{6L+4})$. We remark the bound does not have any explicit dependence on $m$. This in itself is not surprising as one intuitively expects the bound not to scale unfavorably with width. Although recent works have derived width independent generalization bounds for two-layer networks (Ji & Telgarsky, 2020; Telgarsky, 2023; Taheri & Thrampoulidis, 2024), we are not aware of any prior work proving width-independent bounds for learning multi-layer networks with GD. As discussed earlier in the introduction, the bound derived in the closely related work (Chen et al., 2021) scales unfavorably with $m$ as it grows at the rate $\sqrt{m/n}$. In fact, the authors refer to deriving width-independent bounds as an open problem in (Chen et al., 2021, Sec 3.1).

### 2.1.2 CONSISTENCY OF GD WITH NOISY DATA

The results of the last section mainly apply to data settings when the network can find the optimal solution within a bounded distance from initialization that depends on the network's width. This setting was specially tailored to the noiseless case where achieving vanishing test loss was possible. We discuss next the case of learning deep nets by noisy data and show that achieving optimal test loss might be feasible in this setting.

**Theorem 2.3** (Test error for noisy data). *Consider the same setup and notation as Theorem 2.1 and assume the width of the network satisfies $m \geq \beta_L^2 n^{3L+3}$ and the step-size satisfies the conditions of the descent lemma. Then, with high probability over initialization the expected test loss at iteration $T = \sqrt{n}$ is bounded as:*

$$\mathbb{E}_{\mathcal{S}}\left[F(w_T)\right] \leq \left(1 + \frac{4}{\sqrt{n}}\right)\left(F(w^\star) + \frac{\rho^{\star 2}}{\eta\sqrt{n}} + \frac{\rho^{\star 2}}{n\sqrt{n}} + \frac{1}{\sqrt{n}}\right), \tag{9}$$

*where $\rho^\star := \|w^\star - w_0\|$ and $w^\star$ is the minimizer of the population loss i.e., $w^\star = \arg\min_{w \in \mathbb{R}^p} F(w)$.*

The result above shows that GD reaches optimal level given that $\sqrt{n} \gg \|w^\star - w_0\|^2$; as for large $n$, it leads to the simplified expression

$$\mathbb{E}_{\mathcal{S}}\left[F(w_T)\right] - F(w^\star) = O\left(\frac{\left\|w^\star - w_0\right\|^2 + F(w^\star)}{\sqrt{n}}\right).$$

Thus, even in non-interpolating regime, GD can still achieve the optimal solution of over-parameterized deep networks.

The bound in Eq. 9 is derived via a stability argument which bounds the generalization gap based on the training performance. However, contrary to the conditions of Theorem 2.1 here we do not have the interpolation condition as $F(w^\star)$ is not vanishing. This comes at the expense of a larger width condition where the width is polynomial in $n$ whereas poly-logarithmic width was sufficient in Theorem 2.1. The condition on early stopping further guarantees that the test loss reach the Bayes error given $n$ is sufficiently large. It is worth noting that the setting above still is operating almost in the NTK regime. This can be verified by the observation that for the bound to be meaningful it should hold that $\|w^\star - w_0\| \ll \sqrt{n} \lesssim m^{O(1/L)}$, as per the theorem's width condition. Finally, we remark that the result of Thm 2.3 nicely connects to recent empirical and theoretical results (Li et al., 2020) which show that with early-stopping, GD can find a good solution for clustered data with label noise.

## 2.2 Overcoming NTK limitations for XOR dataset

Next, we show the previous NTK guarantees can be improved by using large step-sizes. We consider the stylized set-up of learning the $d-$dimensional XOR data distribution. Consider one-hidden layer network with quadratic activation where $x \in \mathbb{R}^d, w_i \in \mathbb{R}^d$:

$$\Phi(w, x) = \frac{1}{2m} \sum_{i=1}^{m} a_i (x^\top w_i)^2.$$

In the above, $a_i \in \{\pm 1\}$ are fixed during training and satisfy $\sum_i a_i = 0$ and only the first layer weights are trained. For initialization of first layer's weights we have $\forall i \leq m, j \leq d : w_{0,ij} \stackrel{\text{iid}}{\sim} N(0, \frac{1}{d})$. The data points $(x, y) \in \{\pm 1\}^d \times \{\pm 1\}$ are uniformly drawn from the resulting $d$-dimensional distribution of $2^d$ points where the labels are determined as $y = x(1) \cdot x(2)$. For the result in the next theorem, we assume the loss function is the linear loss $f(t) = -t$ and consider mini-batch SGD with the update rule $w^{t+1} = w^t - \eta \widehat{F}(w^t)$, where at each step, $n$ data points are drawn i.i.d. from the distribution to form $\widehat{F}(\cdot)$. The next theorem shows the computational and sample complexities of learning this dataset for the aforementioned setup.

**Theorem 2.4** (Improved guarantees for learning XOR). *Assume mini-batch SGD with batch-size $n \geq d \cdot \log^{14}(d)$ and step-size $\eta = m$. Then, after $T = \log(d)$ iterations the test accuracy satisfies with probability at least $1 - e^{\log(m) - \log^2(d)} - e^{-\frac{m}{16}} - o_d(1)$ over the randomness of initialization and data sampling,*

$$\mathbb{E}_{x,y}\left[\mathbf{1}_{\{y\Phi(w^T, x) > 0\}}\right] \geq 1 - o_d(1). \tag{10}$$

Hence, with logarithmic number of SGD steps we use $\widetilde{O}(d)$ samples in total to reach almost perfect test accuracy. We remark based on the guarantees of Theorem 2.4, the network's width can be constant and at most must be polynomial in $d$ in order for Eq. 10 to hold with high probability. This aligns with our experiments in Fig. 4, demonstrating that the network's width can be independent of data dimension as a network of constant width (where $m = 20$) suffices for learning arbitrary high-dimensional XOR data. We also note the considerable gains in iteration, sample and computational complexities by Theorem 2.4 resulting from escaping the NTK regime with our step-size selection. Contrary to Theorem 2.1 which required small step-size in accordance with the descent lemma, here the step-size is proportional to width. For a comparison with recent results for this dataset we refer to Table 2. To the best of our knowledge these are the best complexities on iteration and network width for this setup.

*Remark* 2.5. The proof of Theorem 2.4 (provided in Appendix E) is nuanced and involves computing expected weights and their corresponding error terms due to SGD sampling noise for each parameter of the network. It is then showed that signal strength (i.e., the strength of important features) grows as $\frac{2^t}{\sqrt{d}}$ whereas the error terms due to sampling noise and initialization grow at most at the rate $(1 + \frac{1}{\text{poly}(\log(d))})^t \frac{\sqrt{d}}{\text{poly}(\log(d))} + \frac{2^t t}{\sqrt{d} \cdot \text{poly}(\log(d))}$. Therefore, after $\log(d)$ steps the noise strength reaches $\frac{\sqrt{d}}{\text{poly}(\log(d))}$ whereas the signal's magnitude is at least $\sqrt{d}$, letting the signal outgrow the noise and leading to the network classifying every point correctly.

## 3 Numerical Results

**Experiments on learning under NTK with small step-size.** In this section, we present numerical results on the behavior of the generalization bound derived in Theorem 2.1 for real-world data (FashionMNIST and MNIST datasets) and compare it with the empirical generalization gap.

For demonstrating our theoretical results, we are interested in the algorithmic-based generalization bound (Eq. 8) derived as,

$$\mathbb{E}_{\mathcal{S}}\left[F(w_T) - \widehat{F}(w_T)\right] \leq \frac{2.2}{n} \mathbb{E}_{\mathcal{S}}\left[\sum_{t=0}^{T-1} \widehat{F}(w_t)\right]. \tag{11}$$

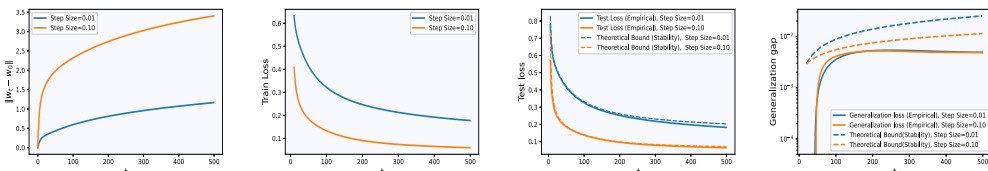

Figure 1: Iteration-based distance from initialization ($\|w_t - w_0\|$), training loss, test loss and generalization gap (i.e., test loss – train loss) for training a two hidden-layer neural network with *FashionMNIST* dataset and two choices of step-size. Here $n = 12 \times 10^3, m = 500$, and total number of parameters $p \approx 6 \times 10^5$.

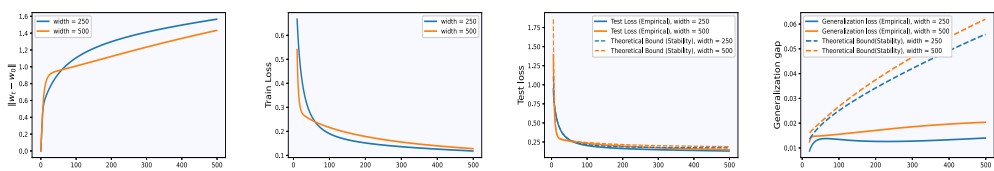

Figure 2: Iteration-based distance from initialization, training loss, test loss and generalization gap for training a two hidden-layer neural network with *FashionMNIST* dataset and $m = 250, 500$. Here $n = 4 \times 10^3, p \approx 2 \times 10^5$ (blue line), $6 \times 10^5$ (red line), and $\eta = 0.02$.

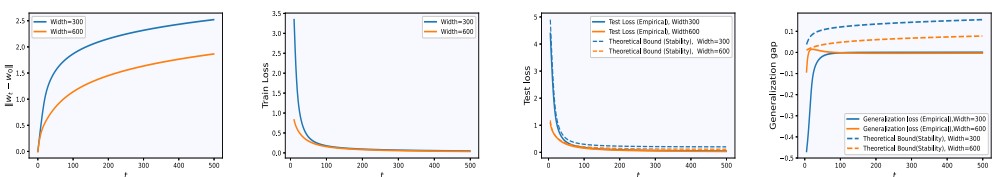

Figure 3: Iteration-based distance from initialization, training loss, test loss and generalization gap for training a two hidden-layer neural network with *MNIST* dataset and $m = 300, 600$. Here $n = 2 \times 10^3, p \approx 3 \times 10^5$ (blue line), $8 \times 10^5$ (red line) and $\eta = 0.02$.

We find it helpful to note that the bound above requires the width condition in the theorem (i.e., $m = \Omega(\|w^\star - w_0\|^{6L+4})$) to be valid. However, verifying this condition is not feasible in general. Moreover, the bound is valid for *expected* generalization gap where the expectation is taken over data sampling. Therefore computing the exact values of both sides of the above inequality is computationally exhaustive. For our experiments we consider one realization to estimate these values. Due to both of these reasons, the theoretical test loss and generalization loss that we present in this section should only be taken as approximations of the general behavior of the bound and not as an actual verified bound on the generalization. However, in order to reduce these impacts we conduct several experiments with varying network's width.

We consider binary classification with a 2-hidden layer network with softplus activation ($\sigma(t) = \log(1 + e^t)$) trained by the logistic loss function. Figure 1 presents train, test and generalization behavior of GD for learning a such a model with FashionMNIST dataset. The two lines in each figure correspond to $\eta = 0.01, 0.1$. In the two rightmost plots, the resulting theoretical generalization and test loss curves derived from Eq. 11 are compared with the empirical values. Note the good alignment between theoretical and empirical behavior for the generalization and test loss.

A similar behavior is observed in Figures 2-3. In Figure 2, we consider a similar setup but we reduce the sample size to $4000$ training data in order to allow larger test-loss behavior. The resulting plots show the training and test loss performance for two choices of $m = 250$ and $m = 500$. In Figure 3, we consider the MNIST dataset for $m = 300$ and $m = 600$ and let the sample size be $n = 2000$. It is

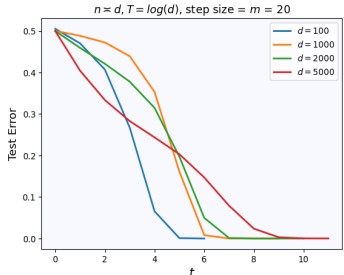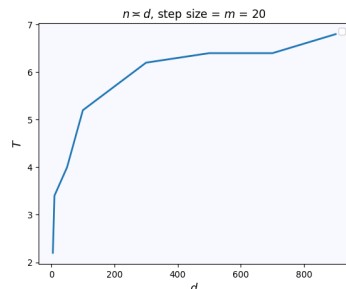

Figure 4: Left: Misclassification error based on iteration in learning the $d-$dimensional XOR distribution with SGD. Right: Total number of SGD steps based on data dimension to reach approximately zero test error.

noteworthy that the findings in both figures bear similarities, with the theoretical bounds providing non-vacuous and accurate approximations for the test loss and generalization gap.

**Experiments on learning the XOR distribution with large step-size.** Figure 4 demonstrates the test error curves associated with learning the XOR distribution according to the setting of Theorem 2.4. In particular, we fix $n = 6d$, $\eta = m = 20$ and set the total number of SGD steps as $T = \lceil \log(d) \rceil$. Note that the number of iterations required to reach perfect accuracy grows with $d$. The right side of Figure 4 provides further insight into the relationship between dimensionality and convergence rate. It displays the total number of SGD steps required to reach a test error below 0.01 for different values of $d$ using $n = 3d$, $m = \eta = 20$. The results are averages over five independent experiments and highlight the logarithmic dependence of the total SGD iterations on data dimension for achieving near-zero test error.

## 4 CONCLUSIONS AND FUTURE DIRECTIONS

We explored the convergence and generalization of smooth neural networks trained with gradient methods. Our first goal in this paper was to derive generalization bounds through a new stability-based approach which had not been discussed in the vast literature of deep learning. These findings represent an improvement over previous results that either required substantial over-parameterization or provided suboptimal generalization rates that depended on the network width. For general noisy data distributions, we also derived generalization guarantees showing that GD can reach the optimal test loss. We also showed that orders of magnitude improvements in sample and computational complexity are possible by surpassing the NTK limitations and using SGD with large step-size. Several directions remain open to future research:

- It remains open to explore whether the minimum width conditions of Theorems 2.1–2.3 can be improved in terms of $\gamma$ or $n$.
- It is also interesting to extend the XOR analysis to the noisy setting where a fraction of the data points have corrupted labels.
- The feature learning phenomenon in multi-index classification tasks remains largely unexplored. While we believe the XOR setup is a must-take first step, extensions to other multi-index models can shed further light on the strengths and limitations of neural networks.
- We also aim to understand the potential benefits of network depth in either the NTK regime or feature learning i.e., whether adding a single layer can improve sample complexity or reduce the total number of SGD iterations.

### ACKNOWLEDGEMENTS

This works was supported by NSF grant CCF 2217058.

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
