APPENDIX

# A  PRELIMINARIES : HESSIAN AND GRADIENT NORM

## A.1  HESSIAN NORM

It is essential to our analysis to obtain Hessian's spectral norm throughout the iterates of GD. With our weights' initialization in place, one can bound the Hessian's spectral norm based on the euclidean distance of the weights from initialization and network's size as in the following assumption.

**Assumption 2** (Hessian's Spectral Norm). The spectral norm of the model's Hessian for any $w \in \mathbb{R}^p, \rho > 0$ such that $\|w - w_0\| \leq \rho$ is bounded as $\|\nabla^2 \Phi(w, \cdot)\|_2 \leq \beta_L \frac{\rho^{3L}}{\sqrt{m}}$ for some constant $\beta_L$ that depends only on $L$.

It can be shown (Liu et al., 2020, Lemma 1) that for standard Gaussian initialization and bounded data, Assumption 2 holds with high-probability over initialization for $\beta_L = e^{O(L)}$.

For the spectral norm of the model's Hessian which is needed by our analysis, we use the following bound which bounds the spectral norm based on the distance from initialization.

**Lemma 1** (Hessian's spectral norm (Liu et al., 2020)). *Assume the defined $L$ layer neural network architecture with standard Gaussian initialization i.e., $w_{0,ij}^{(\ell)} \overset{iid}{\sim} N(0,1)$. Let $\|w - w_0\| \leq \rho$. Then with high probability*

$$\|\nabla^2 \Phi(w, x)\|_2 \leq \beta_L \frac{\rho^{3L}}{\sqrt{m}}$$

*for some constant $\beta_L$ that only depends on $L$.*

As mentioned in the main body, the result above guarantees that Assumption 2 holds with high probability.

## A.2  GRADIENT NORM AT INITIALIZATION

Recall the model for Theorems 2.1-2.3,

$$\Phi(w, x) := \frac{1}{\sqrt{m}} \langle W^{(L+1)}, x^{(L)} \rangle$$

$$x_i^{(\ell+1)} := \frac{1}{\sqrt{m}} \sigma(\langle W_i^{(\ell+1)}, x^{(\ell)} \rangle) \quad \forall \ell \in \{1, \cdots, L-1\}, \forall i \in \{1, \cdots, m\}$$

$$x_i^{(1)} := \sigma(\langle x, W_i^{(1)} \rangle) \quad \forall i \in \{1, \cdots, m\}$$

$$(12)$$

Let us consider the standard Gaussian initialization. We have

$$\frac{\partial \Phi(w, x)}{\partial W_{ij}^{(\ell+1)}} = \frac{1}{\sqrt{m}} x_j^{(\ell)} \sigma'(\langle W_i^{(\ell+1)}, x^{(\ell)} \rangle) \frac{\partial \Phi(w, x)}{\partial x_i^{(\ell+1)}} \tag{13}$$

Therefore,

$$\|\frac{\partial \Phi(w, x)}{\partial W^{(\ell+1)}}\|^2 = \sum_{ij} \|\frac{\partial \Phi(w, x)}{\partial W_{ij}^{(\ell+1)}}\|^2 \leq \frac{1}{m} \|x^{(\ell)}\|^2 \|\frac{\partial \Phi(w, x)}{\partial x^{(\ell+1)}}\|^2 \tag{14}$$

by (Liu et al., 2020, Lemmas F.3 and F.5) :

$$\|x^{(\ell)}\| \leq c_0^\ell \sqrt{m}, \quad \|\frac{\partial \Phi}{\partial x^{(\ell)}}\| \leq c_0^{L-\ell+1} \tag{15}$$

Therefore $\|\frac{\partial \Phi(w_0, x)}{\partial W^{(\ell+1)}}\|^2 \leq c_0^{2L}$ and

$$\|\nabla \Phi(w_0, x)\| = \left(\sum_{\ell=0}^{L} \|\frac{\partial \Phi(w_0, x)}{\partial W^{(\ell+1)}}\|^2\right)^{1/2} \leq \sqrt{L+1} c_0^L.$$

where $c_0$ is any constant that bounds the spectral norm of initialization weights with high probability, i.e., $\frac{\|W_0\|_2}{\sqrt{m}} \leq c_0$. Specially, well-known bounds on the spectral norm of Gaussian matrices guarantee that $c_0$ is constant with high proability (Vershynin, 2018).

Moreover, based on (Liu et al., 2020), if $\|w - w_0\| = \rho$ the spectral norm is bounded by,

$$\|\nabla^2 \Phi(w, x)\|_2 \leq \beta_L \frac{\rho^{3L}}{\sqrt{m}} \tag{16}$$

where $\beta_L$ is a depth-only dependent variable derived to be $\beta_L = e^{O(L)}$.

Now, we use the above relation on Hessian to obtain a bound on the gradient:

$$\frac{\|\nabla \Phi(w^*, x) - \nabla \Phi(w_0, x)\|}{\|w^* - w_0\|} \leq \max_{w \in [w_0, w^*]} \|\nabla^2 \Phi(w, x)\|_2 \tag{17}$$

therefore if $w^* \in \mathcal{B}_\rho(w_0)$,

$$\|\nabla \Phi(w^*, x) - \nabla \Phi(w_0, x)\| \leq \frac{\beta_L \rho^{3L+1}}{\sqrt{m}}. \tag{18}$$

Thus,

$$\|\nabla \Phi(w^*, x)\| - \|\nabla \Phi(w_0, x)\| \leq \|\nabla \Phi(w^*, x) - \nabla \Phi(w_0, x)\| \leq \frac{\beta_L \rho^{3L+1}}{\sqrt{m}}. \tag{19}$$

Thus w.h.p.

$$\max_{i \in [n]} \|\nabla \Phi(w^*, x_i)\| \leq \max_{i \in [n]} \|\nabla \Phi(w_0, x_i)\| + \frac{\beta_L \rho^{3L+1}}{\sqrt{m}} \tag{20}$$

Let us denote the gradient and hessian bounds on $\Phi$ with $C_1$ and $C_2$, respectively. Note that based on our derivations above it holds that $C_1 \leq \frac{\beta_L \rho^{3L+1}}{\sqrt{m}} + G_0$ and $C_2 \leq \frac{\beta_L \rho^{3L}}{\sqrt{m}}$, where $G_0$ is the Lipschtiz parameter at initialization. With these parameters we can now obtain the Lipschitz and Smoothness parameter as well as the smallest eigenvalue of the objective along the gradient descent path. This is derived in the next lemma.

### A.3 OBJECTIVE'S GRADIENT AND HESSIAN ALONG THE GD PATH

**Lemma 2.** *The following are true for the loss gradient and Hessian:*

1. *$\|\nabla \widehat{F}(w)\| \leq C_1 \widehat{F}(w)$.*

2. *$\|\nabla^2 \widehat{F}(w)\|_2 \leq (C_2 + (C_1)^2)\widehat{F}(w)$.*

3. *$\lambda_{\min}\left(\nabla^2 \widehat{F}(w)\right) \geq -C_2 \widehat{F}(w)$.*

*where we define $C_1 := G_0 + \beta_L \|w - w_0\|^{3L+1}/\sqrt{m}$, $C_2 := \beta_L \|w - w_0\|^{3L}/\sqrt{m}$ and $G_0$ is the Lipschitz parameter of the network at initialization i.e., $\|\nabla \Phi(w_0, \cdot)\| \leq G_0$.*

*Proof.* The loss gradient is derived as follows,

$$\nabla \widehat{F}(w) = \frac{1}{n} \sum_{i=1}^{n} f'(y_i \Phi(w, x_i)) y_i \nabla \Phi(w, x_i)$$

Recalling that $y_i \in \{\pm 1\}$, we can write

$$\left\|\nabla\widehat{F}(w)\right\| = \frac{1}{n}\left\|\sum_{i=1}^{n} f'(y_i\Phi(w,x_i))y_i\nabla\Phi(w,x_i)\right\|$$

$$\leq \frac{1}{n}\sum_{i=1}^{n}|f'(y_i\Phi(w,x_i))|\left\|\nabla\Phi(w,x_i)\right\|.$$

$$\leq C_1\widehat{F}(w). \tag{21}$$

For the Hessian of loss, note that

$$\nabla^2\widehat{F}(w) = \frac{1}{n}\sum_{i=1}^{n} f''(y_i\Phi(w,x_i))\nabla\Phi(w,x_i)\nabla\Phi(w,x_i)^\top + f'(y_i\Phi(w,x_i))y_i\nabla^2\Phi(w,x_i). \tag{22}$$

It follows that

$$\left\|\nabla^2\widehat{F}(w)\right\|_2 = \left\|\frac{1}{n}\sum_{i=1}^{n} f'(y_i\Phi(w,x_i))y_i\nabla^2\Phi(w,x_i) + f''(y_i\Phi(w,x_i))\nabla\Phi(w,x_i)\nabla_1\Phi(w,x_i)^\top\right\|$$

$$\leq \frac{1}{n}\sum_{i=1}^{n}|f'(y_i\Phi(w,x_i))|\left\|\nabla^2\Phi(w,x_i)\right\| + |f''(y_i\Phi(w,x_i))|\left\|\nabla\Phi(w,x_i)\nabla\Phi(w,x_i)^\top\right\|$$

$$\leq \frac{1}{n}\sum_{i=1}^{n}|f'(y_i\Phi(w,x_i))|\left\|\nabla^2\Phi(w,x_i)\right\| + |f''(y_i\Phi(w,x_i))|\left\|\nabla\Phi(w,x_i)\right\|^2$$

$$\leq C_2\widehat{F}(w) + (C_1)^2\widehat{F}(w). \tag{23}$$

To lower-bound the minimum eigenvalue of Hessian, note that the loss function $f$ is convex and thus $f''(\cdot) \geq 0$. Therefore the first term in Eq. 22 is positive semi-definite and the second term can be lower-bounded as follows,

$$\lambda_{\min}(\nabla^2\widehat{F}(w)) \geq -\left\|\frac{1}{n}\sum_{i=1}^{n} y_i f'(y_i\Phi(w,x_i))\nabla^2\Phi(w,x_i)\right\|$$

$$\geq -\frac{1}{n}\sum_{i=1}^{n}|y_i f'(y_i\Phi(w,x_i))|\left\|\nabla^2\Phi(w,x_i)\right\|$$

$$\geq -C_2\widehat{F}(w).$$

$\square$

# B  PROOF OF THEOREM 2.1

With the bounds on Hessian and Gradient from the last section, we are ready to prove Theorem 2.1 for obtaining the training and generalization rates of deep nets. We start with proving the training guarantees in the following theorem.

## B.1  PROOF FOR TRAINING LOSS

**Theorem B.1** (Convergence of GD in deep nets). *Let the descent lemma hold. Fix a $T$ and assume the target weights vector $w^\star \in \mathbb{R}^p$ that obtain small training loss such that*

$$\rho^\star \geq \max\left\{\sqrt{\eta T\widehat{F}(w^\star)}, \sqrt{\eta\widehat{F}(w_0)}\right\}.$$

*where $\rho^\star := \|w^\star - w_0\|$. Moreover, assume the width $m$ is large enough such that it satisfies,*

$$m \geq 4\beta_L^2(3\rho^\star)^{6L+4}.$$

*The the training loss satisfies with high probability over initialization,*

$$\frac{1}{T}\sum_{t=1}^{T}\widehat{F}(w_t) \leq 2\widehat{F}(w^\star) + \frac{2\rho^{\star 2}}{\eta T}.$$

To begin the proof, we first state our descent lemma.

**Lemma 3** (Descent lemma). *Consider the notation as in Lemma 2. Assume for all $t \in [T]$ we have $\eta \leq 1/((C_1)^2 + C_2)$, then*

$$\widehat{F}(w_{t+1}) \leq \widehat{F}(w_t) - \frac{\eta}{2}\left\|\nabla\widehat{F}(w_t)\right\|^2.$$

*Proof.* By Taylor expansion and the bound on smoothness parameter of the objective, we immediately deduce:

$$\widehat{F}(w_{t+1}) \leq \widehat{F}(w_t) - \eta\left\|\nabla\widehat{F}(w_t)\right\|^2 + \eta^2\frac{(C_1)^2 + C_2}{2}\left\|\nabla\widehat{F}(w_t)\right\|^2$$

$$\leq \widehat{F}(w_t) - \frac{\eta}{2}\left\|\nabla\widehat{F}(w_t)\right\|^2$$

This completes the proof of the lemma. $\qquad\square$

Define $\rho(t) := \|w_t - w_0\|$ and $\rho^* = \|w^\star - w_0\|$. Also, define $\widetilde{\rho}$ as any constant such that

$$\tilde{\rho} > \max\{\rho(T-1), \rho(T-2), \cdots, \rho(1), \rho^\star\}.$$

Then for any $w \in [w_t, w^\star]$ such that $w = \alpha w_t + (1-\alpha)w^\star$ it holds that

$$\|w - w_0\| \leq \|\alpha w_t + (1-\alpha)w^\star - w_0\| \leq \alpha\|w_t - w_0\| + (1-\alpha)\|w^\star - w_0\| \leq \tilde{\rho},$$

and thus for any $w \in [w_t, w^\star]$

$$\lambda_{\min}\left(\nabla^2\widehat{F}(w)\right) \geq -\widetilde{C}\widehat{F}(w) \tag{24}$$

where $\widetilde{C} := \frac{\beta_L\tilde{\rho}^{3L}}{\sqrt{m}}$, i.e., the smoothness parameter with respect to the distance $\tilde{\rho}$ and as computed by Lemma 2. Therefore, by Taylor remainder theorem, for any fixed $w^\star$ there exists $w'_t \in [w_t, w^\star]$ such that .

$$\widehat{F}(w^\star) \geq \widehat{F}(w_t) + \left\langle\nabla\widehat{F}(w_t), w^\star - w_t\right\rangle + \frac{1}{2}\lambda_{\min}\left(\nabla^2\widehat{F}(w'_t)\right)\|w^\star - w_t\|^2$$

$$\geq \widehat{F}(w_t) + \left\langle\nabla\widehat{F}(w_t), w^\star - w_t\right\rangle - \frac{1}{2}\frac{\beta_L\tilde{\rho}^{3L}}{\sqrt{m}}\widehat{F}(w'_t)\|w^\star - w_t\|^2.$$

As a direct result of the above inequality, we have derived that for any $w^\star$,

$$\widehat{F}(w^\star) \geq \widehat{F}(w_t) + \left\langle\nabla\widehat{F}(w_t), w^\star - w_t\right\rangle - \frac{1}{2}\frac{\beta_L\tilde{\rho}^{3L}}{\sqrt{m}}\max_{w'_t\in[w_t,w^\star]}\widehat{F}(w'_t)\|w - w_t\|^2.$$

Putting this in the relation from descent lemma (Lemma 3) followed by completion of squares using $w_{t+1} - w_t = -\eta\nabla\widehat{F}(w_t)$ gives

$$\widehat{F}(w_{t+1}) \leq \widehat{F}(w^\star) - \left\langle\nabla\widehat{F}(w_t), w^\star - w_t\right\rangle - \frac{\eta}{2}\left\|\nabla\widehat{F}(w_t)\right\|^2 + \frac{1}{2}\frac{\beta_L\tilde{\rho}^{3L}}{\sqrt{m}}\max_{w'_t\in[w_t,w^\star]}\widehat{F}(w'_t)\|w^\star - w_t\|^2$$

$$= \widehat{F}(w^\star) + \frac{1}{\eta}\left(\|w^\star - w_t\|^2 - \|w^\star - w_{t+1}\|^2\right) + \frac{1}{2}\frac{\beta_L\tilde{\rho}^{3L}}{\sqrt{m}}\max_{w'_t\in[w_t,w^\star]}\widehat{F}(w'_t)\|w^\star - w_t\|^2. \tag{25}$$

Telescoping the above display for $t = 0, \ldots, T-1$, we deduce that,

$$\frac{1}{T}\sum_{t=1}^{T}\widehat{F}(w_t) \leq \widehat{F}(w^\star) + \frac{\|w^\star - w_0\|^2}{\eta T} + \frac{\beta_L\tilde{\rho}^{3L}}{2T}\sum_{t=0}^{T-1}\max_{w'_t\in[w_t,w^\star]}\widehat{F}(w'_t)\|w^\star - w_t\|^2. \tag{26}$$

In order to proceed with the above relation, it is necessary to further simplify the terms $\max_{w'_t\in[w_t,w^\star]}\widehat{F}(w'_t)$. In doing so, we utilize the following Generalized Local Quasi-convexity (GLQC) property from Taheri & Thrampoulidis (2024).

**Proposition 1** (Prop. 8 in Taheri & Thrampoulidis (2024))**.** *Let $w_1, w_2 \in \mathbb{R}^p$ be two arbitrary points. Suppose $\widehat{F} : \mathbb{R}^p \to \mathbb{R}$ satisfies the self-bounded weak convexity property with parameter $\kappa$ along the line $[w_1, w_2]$. Assume $\|w_1 - w_2\| \leq D$ where $D < \sqrt{2/\kappa}$. Then,*

$$\max_{v \in [w_1, w_2]} \widehat{F}(v) \leq \tau \cdot \max\{\widehat{F}(w_1), \widehat{F}(w_2)\}. \tag{27}$$

*where $\tau := \left(1 - \kappa D^2/2\right)^{-1}$.*

For our objective we have the minimum eigenvalue of hessian being lower bounded by $\kappa = \widetilde{C} := \beta_L \tilde{\rho}^{3L}$. Therefore for $w_t, w^\star$, if $\|w^\star - w_t\|^2 < D^2 < 1/2\widetilde{C}$, then the GLQC property holds with $\tau = 4/3$. Recall that by the defnition of $\tilde{\rho}$ it holds that $\|w^\star - w_t\| \leq \tilde{\rho}$. Thus, to guarantee the conditions of the proposition above are satisfied it suffices to hold that

$$\sqrt{m} \geq 2\beta_L \tilde{\rho}^{3L+2}. \tag{28}$$

which is exactly the width condition of our theorem. With this condition on width and the resulting simplifications from the proposition, we can write:

$$\frac{1}{T} \sum_{t=1}^{T} \widehat{F}(w_t) \leq \widehat{F}(w^\star) + \frac{\|w^\star - w_0\|^2}{\eta T} + \frac{\beta_L \tilde{\rho}^{3L}}{\sqrt{m}} \frac{2}{3T} \sum_{t=0}^{T-1} \max\{\widehat{F}(w^\star), \widehat{F}(w_t)\} \|w^\star - w_t\|^2,$$

$$\leq \widehat{F}(w^\star) + \frac{\|w^\star - w_0\|^2}{\eta T} + \frac{1}{3T} \sum_{t=0}^{T-1} \max\{\widehat{F}(w^\star), \widehat{F}(w_t)\}$$

$$\leq \frac{4}{3} \widehat{F}(w^\star) + \frac{\|w^\star - w_0\|^2}{\eta T} + \frac{1}{3T} \sum_{t=0}^{T-1} \widehat{F}(w_t)$$

Thus,

$$\frac{1}{T} \sum_{t=1}^{T} \widehat{F}(w_t) \leq 2\widehat{F}(w^\star) + \frac{3\|w^\star - w_0\|^2}{2\eta T} + \frac{1}{2T} \widehat{F}(w_0) \tag{29}$$

Thus far, we have characterized the convergence under the condition that $\sqrt{m} \geq 2\beta_L \tilde{\rho}^{3L+2} = 2\beta_L(\max\{\|w_{T-1} - w_0\|, \cdots, \|w_1 - w_0\|, \|w^\star - w_0\|\})^{3L+2}$. It remains to characterize the above condition on width independent of the iteration number. In fact, in what follows we are able to show it is sufficient to reduce this condition such that it depends on only $\|w^\star - w_0\|$. We do it by an iterate-wise norm bound on $\|w^\star - w_t\|$ based on $\|w^\star - w_0\|$.

Recall that we had,

$$\widehat{F}(w_{t+1}) \leq \widehat{F}(w^\star) + \frac{1}{\eta}\left(\|w^\star - w_t\|^2 - \|w^\star - w_{t+1}\|^2\right) + \frac{1}{2}\widetilde{C} \max_{w_t' \in [w_t, w^\star]} \widehat{F}(w_t') \|w^\star - w_t\|^2. \tag{30}$$

where recall that $\widetilde{C} = \frac{\beta_L \tilde{\rho}^{3L}}{\sqrt{m}}$, and $\tilde{\rho} := \max\{\rho(1), \rho(2), ...\rho(T-1), \rho^\star\}$.

Hence,

$$\|w_{t+1} - w^\star\|^2 \leq \|w_t - w^\star\|^2 + \eta\widehat{F}(w^\star) + \frac{\eta}{2}\frac{\beta_L \tilde{\rho}^{3L}}{\sqrt{m}} \max_{w_t' \in [w_t, w^\star]} \widehat{F}(w_t') \|w^\star - w_t\|^2. \tag{31}$$

The following lemma proves through an induction-based argument that choosing $\tilde{\rho} = 3\rho^\star$ and the corresponding width condition based on this $\tilde{\rho}$ is sufficient to ensure that $\rho(t) \leq \tilde{\rho}$ for all iterates $t \in [T]$; which in turn guarantees the above argument for convergence to be valid.

**Lemma 4.** *Fix any $T \geq 0$ and assume any $w^\star$ such that*

$$\|w^\star - w_0\|^2 \geq \max\{\eta T \widehat{F}(w^\star), \eta\widehat{F}(w_0)\}. \tag{32}$$

*Pick $\tilde{\rho} = 3\rho^\star = 3\|w^\star - w_0\|$ and assume $m$ is large enough such that $\tilde{\rho}^2 < 1/2\widetilde{C}$ where $\widetilde{C} = \frac{\beta_L \tilde{\rho}^{3L}}{\sqrt{m}}$. Then $\rho(t) := \|w_t - w_0\| \leq \tilde{\rho}$ and $\|w_t - w^\star\| \leq \tilde{\rho}$ for all $t \in [T]$.*

*Proof.* The statements of the lemma are correct for $t = 0$. Now assume for $t \leq T - 1$ we have $\rho(t) \leq \widetilde{\rho}$ and $\|w_t - w^\star\| \leq \widetilde{\rho}$. Recall, by 31

$$\|w_T - w^\star\|^2 \leq \|w_{T-1} - w^\star\|^2 + \eta\widehat{F}(w^\star) + \frac{\eta}{2}\frac{\beta_L\widetilde{\rho}^{3L}}{\sqrt{m}}\|w_{T-1} - w^\star\|^2 \max_{w'_{T-1} \in [w_{T-1}, w^\star]} \widehat{F}(w'_{T-1}) \tag{33}$$

Note that by the assumption of the lemma and the induction assumption it holds that
$$\max\{\|w_{T-1} - w_0\|, \|w^\star - w_0\|\} \leq \widetilde{\rho}.$$
As a result, for any $w \in [w_{T-1}, w^\star]$, it holds that $\|w - w^\star\| \leq \widetilde{\rho}$ and thus $\|\nabla\widehat{F}(w)\| \geq \frac{\beta_L\widetilde{\rho}^{3L}}{\sqrt{m}}\widehat{F}(w)$. Moreover, note that as per the lemma's assumption $\|w_{T-1} - w^\star\|^2 \leq \widetilde{\rho}^2 \leq 1/2\widetilde{C}$. Thus, the conditions of Proposition 1 hold with $\tau = 4/3$ and we have

$$\|w_T - w^\star\|^2 \leq \|w_{T-1} - w^\star\|^2 + \eta\widehat{F}(w^\star) + \frac{2\eta}{3}\frac{\beta_L\widetilde{\rho}^{3L}}{\sqrt{m}}\|w_{T-1} - w^\star\|^2 \cdot \max\{\widehat{F}(w_{T-1}), \widehat{F}(w^\star)\}$$

Noting again that $\|w_{T-1} - w^\star\|^2 \leq \widetilde{\rho}^2 \leq 1/2\widetilde{C}$ yields,

$$\|w_T - w^\star\|^2 \leq \|w_{T-1} - w^\star\|^2 + \eta\widehat{F}(w^\star) + \frac{\eta}{3}\max\{\widehat{F}(w_{T-1}), \widehat{F}(w^\star)\}$$

$$\leq \|w_{T-1} - w^\star\|^2 + \frac{4\eta}{3}\widehat{F}(w^\star) + \frac{\eta}{3}\widehat{F}(w_{T-1})$$

Since the conditions of the lemma hold for all $t \in [T - 1]$, we can telescope the summation to obtain,

$$\|w_T - w^\star\|^2 \leq \|w_0 - w^\star\|^2 + \frac{4\eta}{3}T\widehat{F}(w^\star) + \frac{\eta}{3}\sum_{t=0}^{T-1}\widehat{F}(w_t)$$

$$\leq \|w_0 - w^\star\|^2 + \frac{4\eta}{3}T\widehat{F}(w^\star) + \frac{\eta}{3}\widehat{F}(w_0) + \frac{\eta}{3}\sum_{t=1}^{T-1}\widehat{F}(w_t)$$

$$\leq \|w_0 - w^\star\|^2 + \frac{4\eta}{3}T\widehat{F}(w^\star) + \frac{\eta}{3}\widehat{F}(w_0) + \frac{\eta}{3}\left(2T\widehat{F}(w^\star) + \frac{3\|w^\star - w_0\|^2}{2\eta} + \frac{1}{2}\widehat{F}(w_0)\right)$$

$$= \|w_0 - w^\star\|^2 + 2\eta T\widehat{F}(w^\star) + \frac{\eta}{2}\widehat{F}(w_0) + \frac{\|w^\star - w_0\|^2}{2}$$

Therefore by the conditions of the lemma and noting that $A_0 = \|w_0 - w^\star\|^2 = (\rho^\star)^2$:

$$\|w_T - w^\star\|^2 \leq \|w_0 - w^\star\|^2 + 2\|w_0 - w^\star\|^2 + \frac{1}{2}\|w^\star - w_0\|^2 + \frac{1}{2}\|w^\star - w_0\|^2 = 4(\rho^\star)^2$$

Thus, $\|w_T - w^\star\|^2 \leq 9(\rho^\star)^2 = \widetilde{\rho}^2$. It remains to prove that $\rho(T) := \|w_T - w_0\| \leq \widetilde{\rho}$. To obtain this, note that by triangle inequality,

$$\|w_T - w_0\| \leq \|w_T - w^\star\| + \|w^\star - w_0\|$$
$$\leq 2\rho^\star + \rho^\star = 3\rho^\star = \widetilde{\rho}.$$

The proof is complete. $\qquad\qquad\qquad\square$

The lemma above shows that controlling $\rho^\star = \|w^\star - w_0\|$ is enough to guarantee that GLQC property (Proposition 1) holds throughout the gradient path. In particular, if the width is large enough such that for $\widetilde{\rho} = 3\rho^\star$: $\widetilde{\rho}^2 \leq \frac{1}{2\widetilde{C}} = \frac{\sqrt{m}}{2\beta_L\widetilde{\rho}^{3L}}$ or equivalently

$$\sqrt{m} \geq 2\beta_L\widetilde{\rho}^{3L+2}.$$

then the conditions of the lemma hold and the bounds on the training loss are valid. Note that we need to ensure the conditions of the descent lemma (Lem 3) hold throughout the GD iterates. This is guaranteed by taking

$$\eta \leq \frac{1}{(\widetilde{C}_1)^2 + \widetilde{C}} \tag{34}$$

where note that $\tilde{C}_1 = G_0 + \beta_L\frac{(3\|w^\star - w_0\|)^{3L+1}}{\sqrt{m}}$, and $\tilde{C} = \beta_L\frac{(3\|w^\star - w_0\|)^{3L}}{\sqrt{m}}$. As per Lemma 2, this condition ensures that step-size is smaller than the inverse of objective's smoothness parameter throughout all iterates since $\|w_t - w_0\| \leq 3\|w^\star - w_0\|$ for all $t \in [T]$.

This completes the proof of Theorem B.1.

## B.2 Proof for Generalization loss

**Theorem B.2** (Generalization loss of GD in deep nets). *Let the descent lemma hold. Fix a $T$ and assume the target weights vector $w^\star \in \mathbb{R}^p$ that separates the data distribution such that*

$$\rho^\star \geq \max\left\{ \sqrt{\eta T \widehat{F}(w^\star)}, \sqrt{\eta \widehat{F}(w_0)} \right\}. \tag{35}$$

*where $\rho^\star := \|w^\star - w_0\|$. Moreover, assume the width $m$ is large enough such that it satisfies,*

$$\sqrt{m} \geq 4\beta_L \bar{\rho}^{3L+2} \tag{36}$$

*where $\bar{\rho} = 6\rho^\star$. Then the expected generalization error satisfies with high probability,*

$$\mathbb{E}_{\mathcal{S}}\left[ F(w_T) - \widehat{F}(w_T) \right] \leq 9 \frac{(G_0 + 1/4)^2 \rho^{\star 2}}{n}, \tag{37}$$

*where $G_0$ is the Lipschitz parameter of network at initialization i.e., $\|\nabla\Phi(w_0, \cdot)\| \leq G_0$.*

As per the last theorem, we know if $m$ is large enough based on $\rho^\star$, then the hessian during the iterates of GD satisfies $\|\nabla^2\Phi(w, x)\| \leq C_2$ and thus as per Lemma 2 we have,

1. $\|\nabla\widehat{F}(w)\|_F \leq C_1 \widehat{F}(w)$.
2. $\|\nabla^2\widehat{F}(w)\|_2 \leq (C_2 + (C_1)^2)\widehat{F}(w)$.
3. $\lambda_{\min}\left(\nabla^2\widehat{F}(w)\right) \geq -C_2\widehat{F}(w)$.

throughout this section and with a slight abuse of notation, we define $C_1 = G_0 + \beta_L(3\|w - w_0\|)^{3L+1}/\sqrt{m}$, $C_2 = \beta_L(3\|w - w_0\|)^{3L}/\sqrt{m}$ and $G_0$ is the Lipschitz parameter of the network at initialization i.e., $\|\nabla\Phi(w_0, \cdot)\| \leq G_0$. Recall that in obtaining these quantities we used the guarantees of Thm 2.1 that along the gradient descent path $\max\{\|w_t - w_0\|, \|w_t - w^\star\|\} \leq 3\|w^\star - w_0\|$

We derive the generalization bound through leave-one-out error estimates. In particular, let $w_t^{\neg i}$ denote the output of $t$-th iteration of GD on the leave-one-out loss $F^{\neg i}(\cdot) : \mathbb{R}^p \to \mathbb{R}$ defined as $F^{\neg i}(w) := \frac{1}{n}\sum_{j \neq i} \widehat{F}_j(w)$. Then, a simple calculation based on the Lipschitz property of the objective relates the generalization loss to the leave-one-out error as stated in the following lemma.

**Lemma 5** (Lem. 6 in Schliserman & Koren (2022), Thm. 2 in Lei & Ying (2020)). *Suppose each sample loss $f(\cdot, z)$ is $G$-Lipschitz for almost surely all data points $z \sim \mathcal{D}$. Then, the following relation holds between expected generalization loss and model stability at any iterate $T$,*

$$\mathbb{E}_{\mathcal{S}}\left[ F(w_T) - \widehat{F}(w_T) \right] \leq 2G \cdot \mathbb{E}_{\mathcal{S}}\left[ \frac{1}{n} \sum_{i=1}^{n} \|w_T - w_T^{\neg i}\| \right]. \tag{38}$$

We will use the fact that under the descent lemma $\eta < 1/(C_2 + C_1^2)$, based on (Taheri & Thrampoulidis, 2024, Lem. 9):

$$\left\| \left( w - \eta\nabla\widehat{F}(w) \right) - \left( w' - \eta\nabla\widehat{F}(w') \right) \right\| \leq \max_{\tilde{w} \in [w, w']} (1 + \eta C_2 \widehat{F}(\tilde{w})) \|w - w'\|. \tag{39}$$

Then, we can bound the leave-one-out error as follows:

$$\left\| w_{t+1} - w_{t+1}^{\neg i} \right\| \leq \left\| \left( w_t - \eta\nabla\widehat{F}^{\neg i}(w_t) \right) - \left( w_t^{\neg i} - \eta\nabla\widehat{F}^{\neg i}(w_t^{\neg i}) \right) \right\| + \frac{\eta}{n} \left\| \nabla\widehat{F}_i(w_t) \right\|$$

$$\leq \left\| \left( w_t - \eta\nabla\widehat{F}^{\neg i}(w_t) \right) - \left( w_t^{\neg i} - \eta\nabla\widehat{F}^{\neg i}(w_t^{\neg i}) \right) \right\| + \frac{\eta C_1}{n} \widehat{F}_i(w_t)$$

$$\leq \left( 1 + \eta C_2 \max_{w_t' \in [w_t, w_t^{\neg i}]} \widehat{F}^{\neg i}(w_t') \right) \|w_t - w_t^{\neg i}\| + \frac{\eta C_1}{n} \widehat{F}_i(w_t),$$

By the aforementioned property from Proposition 1, we have:

$$\max_{w'_t \in [w_t, w_t^{\neg i}]} \widehat{F}^{\neg i}(w'_t) \leq \frac{4}{3} \max\{\widehat{F}^{\neg i}(w_t), \widehat{F}^{\neg i}(w_t^{\neg i})\}$$

$$\leq \frac{4}{3} \max\{\widehat{F}(w_t), \widehat{F}^{\neg i}(w_t^{\neg i})\}$$

$$\leq \frac{4}{3}(\widehat{F}(w_t) + \widehat{F}^{\neg i}(w_t^{\neg i}))$$

Note that for $w'_t$ we have

$$\|w'_t - w_0\| \leq \alpha\|w_t - w_0\| + (1-\alpha)\|w_t^{\neg i} - w_0\| \leq \widetilde{\rho} \tag{40}$$

therefore the self-bounded weak convexity holds with $\kappa = C_2$. Moreover,

$$\|w_t - w_t^{\neg i}\| \leq \|w_t - w_0\| + \|w_t^{\neg i} - w_0\| \leq 3\rho^\star + 3\rho^\star = 6\rho^\star = 2\widetilde{\rho}$$

Denote $\bar{\rho} := 6\rho^\star (= 2\widetilde{\rho})$. If $m$ is large enough such that $\bar{\rho}^2 < 1/2\bar{C}_2$ where recall $\bar{C}_2 = \frac{\beta_L \bar{\rho}^{3L}}{\sqrt{m}}$, then the GLQC property holds with $\tau = 4/3$ and we have,

$$\|w_{t+1} - w_{t+1}^{\neg i}\| \leq \left(1 + \frac{4}{3}\eta\bar{C}_2(\widehat{F}(w_t) + \widehat{F}^{\neg i}(w_t^{\neg i}))\right) + \frac{\eta C_1}{n}\widehat{F}_i(w_t)$$

$$\leq \exp(\frac{4}{3}\eta\bar{C}_2(\widehat{F}(w_t) + \widehat{F}^{\neg i}(w_t^{\neg i}))) + \frac{\eta C_1}{n}\widehat{F}_i(w_t)$$

$$\leq \frac{\eta C_1}{n}\sum_{k=0}^{t}\exp\left(\sum_{\tau=1}^{t}\frac{4}{3}\eta\bar{C}_2(\widehat{F}(w_\tau) + \widehat{F}^{\neg i}(w_\tau^{\neg i}))\right)\widehat{F}_i(w_k)$$

Fix a $T$ and assume $\bar{C}_2$ is small enough such that

$$\frac{4}{3}\eta\bar{C}_2\sum_{\tau=1}^{T}(\widehat{F}(w_\tau) + \widehat{F}^{\neg i}(w_\tau^{\neg i})) \leq \frac{16}{3}\bar{C}_2(\eta T\widehat{F}(w^\star) + \rho^{\star 2}) \leq 0.05 \tag{41}$$

then,

$$\|w_T - w_T^{\neg i}\| \leq \frac{\eta C_1 \exp(0.05)}{n}\sum_{t=0}^{T}\widehat{F}_i(w_t)$$

$$\leq 1.1\frac{\eta C_1}{n}\sum_{t=0}^{T}\widehat{F}_i(w_t).$$

The choice of 0.05 in Eq. 41 is to ensure that the resulting constant in the generalization bound is close to 1. By averaging over $i \in [n]$ and by our training loss guarantees from Theorem 2.1:

$$\frac{1}{n}\sum_{i=1}^{n}\|w_T - w_T^{\neg i}\| \leq 1.1\frac{\eta C_1}{n}\sum_{t=0}^{T}\widehat{F}(w_t)$$

$$\leq 2.2\frac{C_1}{n}(\eta T\widehat{F}(w^\star) + \rho^{\star 2})$$

based on Lemma 5 and noting that $G = C_1$, we find that

$$\mathbb{E}_{\mathcal{S}}\left[F(w_T) - \widehat{F}(w_T)\right] \leq \frac{4.4}{n}\mathbb{E}_{\mathcal{S}}\left[C_1^2\eta T\widehat{F}(w^\star) + C_1^2\rho^{\star 2}\right] \tag{42}$$

Recall from Lemma 2 that $C_1 \leq G_0 + \beta_L\|w_t - w_0\|^{3L+1}/\sqrt{m}$ for all $t \in [T]$. As we require $\sqrt{m} \geq 4\beta_L(6\|w^\star - w_0\|^{3L+2})$ and from convergence guarantees it holds that $\|w_t - w_0\| \leq 3\|w^\star - w_0\|$ we conclude that

$$C_1 \leq G_0 + \frac{1}{2^{3L+2}\|w^\star - w_0\|} \leq G_0 + 1/4,$$

where in the last step we assumed without loss of generality that $\|w^\star - w_0\| \geq 1$. This completes the proof for the generalization error in Theorem B.2. The proof of Theorem 2.1 is complete.

## C  PROOF OF COROLLARY 1

**Corollary 2** (Restatement of Corollary 1). *Assume the tangent kernel of the model at initialization separates the data with margin $\gamma$ i.e., for a unit-norm $w \in \mathbb{R}^p$ it holds for all $i \in [n]$ : $y_i \langle \nabla \Phi(w_0, x_i), w \rangle \geq \gamma$. Define constant $B > 0$ that bounds the model's output at initialization i.e., $\forall i \in [n] : |\Phi(w_0, x_i)| < B$. Assume the width is large enough such that $m \geq \beta_L^2 (\frac{2B + \log(1/\epsilon)}{\gamma})^{6L+4}$. Then there exists $w^\star \in \mathbb{R}^p$ such that $F(w^\star) \leq \epsilon$ and $w^\star$ lies at a Euclidean distance of order $O(1/\gamma)$ from initialization where $\|w^\star - w_0\| \leq \frac{1}{\gamma}(2B + \log(1/\epsilon))$.*

*Proof of Corollary 1.* By Taylor there exists $w' \in [w^\star, w_0]$ such that,

$$y_i \Phi(w^\star, x_i) = y_i \Phi(w_0, x_i) + y_i \langle \nabla \Phi(w_0, x_i), w^\star - w_0 \rangle + \frac{1}{2} y_i \langle w^\star - w_0, \nabla^2 \Phi(w', x_i)(w^\star - w_0) \rangle$$

.

Pick $w^\star = w_0 + \frac{w}{\gamma}(2B + \log(1/\varepsilon))$. Since $\|w\| = 1$, we derive the desired for $\|w^\star - w_0\|$. we have $\|\nabla^2 \Phi(w', x_i)\| \leq O(\frac{\rho^{3L}}{\sqrt{m}})$. Choosing $\rho = \|w^\star - w_0\| = \frac{1}{\gamma}(2B + \log(1/\epsilon))$ and noting the given condition on $m$ implies $m \geq \beta_L^2 \rho^{6L+4}$ we obtain $\frac{\beta_L \rho^{3L+2}}{\sqrt{m}} \leq 1$. Therefore with the given assumption on $m \geq \frac{\beta_L^2}{\gamma^{6L+4}}(2B + \log(1/\epsilon))^{6L+4}$ leads to $\frac{1}{2} \|\nabla^2 \Phi(w', x_i)\| \|w^\star - w_0\|^2 \leq 1$. Then,

$$y_i \Phi(w^\star, x_i) \geq -|y_i \Phi(w_0, x_i)| + y_i \langle \nabla_1 \Phi(w_0, x_i), w^\star - w_0 \rangle - \frac{1}{2} \|\nabla_1^2 \Phi(w', x_i)\| \|w^\star - w_0\|^2$$
$$\geq -B + 2B + \log(1/\varepsilon) - B$$
$$= \log(1/\varepsilon)$$

where we assumed without loss of generality that $B \geq 1$. After applying the logistic loss function on $y_i \Phi(w^\star, x_i)$, the inequality above gives $\widehat{F}(w) \leq \varepsilon$. This completes the proof. $\qquad \square$

## D  CONSISTENCY OF GD IN DEEP NETS: PROOF OF THEOREM 2.3

**Theorem D.1** (Consistency). *Assume the width of the network satisfies $m \geq \beta_L^2 n^{3L+3}$ and the step-size satisfies $\eta < 1/L_F$. Then with high probability the expected test loss at iteration $T \geq \sqrt{n}$ is bounded as:*

$$\mathbb{E}_S[F(w_T)] \leq \left(1 + \frac{4}{\sqrt{n}}\right)\left(F(w^\star) + \frac{\rho^{\star 2}}{\eta \sqrt{n}} + \frac{\rho^{\star 2}}{n \sqrt{n}} + \frac{1}{\sqrt{n}}\right) \tag{43}$$

*where $w^\star$ is the minimizer of the population loss i.e., $w^\star = \arg\min_{w \in \mathbb{R}^p} F(w)$.*

Recall that based on 39

$$\left\|(w - \eta \nabla \widehat{F}(w)) - \left(w' - \eta \nabla \widehat{F}(w')\right)\right\| \leq 1 + \eta \max_\alpha C_2 \|w - w'\|, \tag{44}$$

where $C_2 := O(\frac{\beta_L \rho^{3L}}{\sqrt{m}})$ and $\rho := \|w - w_0\|$

In particular, note that for GD, we can obtain for $\|w - w_0\|$:

$$\|w_T - w_0\| \leq \eta \sum_{t=0}^{T-1} \|\nabla \widehat{F}(w_t)\|$$
$$\leq \eta \sum_{t=0}^{T-1} C_1$$
$$= \eta T C_1, \tag{45}$$

where the first step follows by update rule of GD and in the second step we used the Lipschitz property of objective during GD iterates that ensures $\|\nabla \widehat{F}(w_t)\|$ remains bounded by $C_1$ for all $t \in [T]$. Thus, we can thus show by induction that $\|w_T - w_0\| \leq \eta T C_1$. Then for $m = \Omega(T^2)$ and $w_T^{\neg i}, w_T$ :

$$\left\|(w_T - \eta \nabla \widehat{F}(w_T)) - \left(w_T^{\neg i} - \eta \nabla \widehat{F}(w_T^{\neg i})\right)\right\| \leq (1 + \eta C_2) \|w_T - w_T^{\neg i}\|, \tag{46}$$

where $C_2 = \beta_L \tilde{\rho}^{3L}/\sqrt{m}$ is the smoothness parameter of the objective throughout the GD updates $w_t$ for all $t$. Then,

$$\left\| w_{t+1} - w_{t+1}^{\neg i} \right\| \leq \left\| \left( w_t - \eta \nabla \widehat{F}^{\neg i}(w_t) \right) - \left( w_t^{\neg i} - \eta \nabla \widehat{F}^{\neg i}(w_t^{\neg i}) \right) \right\| + \frac{\eta}{n} \left\| \nabla \widehat{F}_i(w_t) \right\|$$

$$\leq \left\| \left( w_t - \eta \nabla \widehat{F}^{\neg i}(w_t) \right) - \left( w_t^{\neg i} - \eta \nabla \widehat{F}^{\neg i}(w_t^{\neg i}) \right) \right\| + \frac{\eta C_1}{n} \widehat{F}_i(w_t)$$

$$\leq (1 + \eta C_2) \left\| w_t - w_t^{\neg i} \right\| + \frac{\eta C_1}{n} \widehat{F}_i(w_t),$$

Recall that in the above $\widehat{F}_i(\cdot)$ refers to the objective at the $i$th data sample and thus has the same Lipschits constant as $\widehat{F}(w) = \frac{1}{n} \sum_{i=1}^{n} \widehat{F}_i(w)$.

Noting that $1 + x \leq e^x$ for $x \geq 0$, we can proceed with the above inequality as follows by unrolling the iterates of $\|w_t - w_t^{\neg i}\|$ and the fact that $w_0 = w_0^{\neg i}$:

$$\left\| w_{t+1} - w_{t+1}^{\neg i} \right\| \leq \exp(\eta C_2)\|w_t - w_t^{\neg i}\| + \frac{\eta C_1}{n} \widehat{F}_i(w_t)$$

$$\leq \frac{\eta C_1}{n} \sum_{k=0}^{t} \exp(\eta C_2 t) \widehat{F}_i(w_k).$$

Recall that, $C_2 = \beta_L \tilde{\rho}^{3L}/\sqrt{m}$, and $\tilde{\rho} = \max\{\|w_t - w_0\|, \|w_{t-1} - w_0\|, \cdots, \|w_1 - w_0\|\}$. Based on Eq. 45, $\|w_t - w_0\| \leq \eta t C_1$ which implies $\tilde{\rho} \leq \eta t C_1$. As a result, for some fixed $T$:

$$\|w_T - w_T^{\neg i}\| \leq \frac{\eta C_1}{n} \sum_{t=0}^{T-1} \exp\left( \frac{\eta \beta_L T}{\sqrt{m}} (\eta T C_1)^{3L} \right) \widehat{F}_i(w_t).$$

Thus for $m$ larger enough such that $\sqrt{m} \geq 2\beta_L (\eta C_1 T)^{3L+1}$, the exponential term is smaller than 2 and we conclude that,

$$\left\| w_T - w_T^{\neg i} \right\| \leq \frac{2\eta C_1}{n} \sum_{t=0}^{T-1} \widehat{F}_i(w_t). \tag{47}$$

Therefore, by averaging over $i \in [n]$ we derive the on-average leave-one-out:

$$\frac{1}{n} \sum_i \left\| w_T - w_T^{\neg i} \right\| \leq \frac{2\eta C_1}{n} \sum_{t=0}^{T-1} \widehat{F}(w_t) \tag{48}$$

The only remaining part is characterising the right hand side of the above inequality. This is done as follows.

### PROOF OF OPTIMIZATION ERROR

Define $\rho(t) := \|w_t - w_0\|$ and $\rho^* = \|w^\star - w_0\|$. if we define $\tilde{\rho} > \max\{\rho(t), \rho^\star\}$ for all $t \in [T]$ then for any $w \in [w_t, w^\star]$ it holds that $\|w - w_0\| \leq \tilde{\rho}$ and thus based on Lemma 2

$$\lambda_{\min}(\nabla^2 \widehat{F}(w)) \geq -\widetilde{C}\widehat{F}(w), \tag{49}$$

where $\widetilde{C} := \frac{\beta_L \tilde{\rho}^{3L}}{\sqrt{m}}$. Therefore by Eq. 26,

$$\widehat{F}(w_T) \leq \frac{1}{T} \sum_{t=1}^{T} \widehat{F}(w_t) \leq \widehat{F}(w^\star) + \frac{\|w^\star - w_0\|^2}{\eta T} + \widetilde{C} \frac{1}{2T} \sum_{t=0}^{T-1} \max_{w_t' \in [w_t, w^\star]} \widehat{F}(w_t') \|w^\star - w_t\|^2,$$

$$\leq \widehat{F}(w^\star) + \frac{\|w^\star - w_0\|^2}{\eta T} + \widetilde{C} \frac{1}{2T} \sum_{t=0}^{T-1} \|w^\star - w_t\|^2$$

$$\leq \widehat{F}(w^\star) + \frac{\|w^\star - w_0\|^2}{\eta T} + \widetilde{C} \|w^\star - w_0\|^2 + \widetilde{C} \frac{1}{T} \sum_{t=0}^{T-1} \|w_t - w_0\|^2.$$

For the last term, we have a simple bound:

$$\|w_t - w_0\| = \eta \|\sum_{k=0}^{t-1} \nabla \widehat{F}(w_k)\| \le \eta \sum_{k=0}^{t-1} \|\nabla \widehat{F}(w_k)\| \le \eta t C_1$$

where $C_1$ denote the lipschitz parameter bound along the GD iterates and the last inequality follows by descent lemma condition.

Then,

$$\widehat{F}(w_T) \le \widehat{F}(w^\star) + \frac{\|w^\star - w_0\|^2}{\eta T} + \widetilde{C} \|w^\star - w_0\|^2 + \eta^2 \widetilde{C} C_1^2 T^2.$$

Pick large enough width such that $m \ge \beta_L^2 n^{3L+3} (\eta C_1)^{6L}$ and choose $T = \sqrt{n}$. Then $\tilde{\rho} < (\eta C_1 n)^{3L/2}$ and

$$\widehat{F}(w_T) \le \widehat{F}(w^\star) + \frac{\rho^{\star 2}}{\eta \sqrt{n}} + \frac{\beta_L \tilde{\rho}^{3L}}{\sqrt{m}} (\rho^{\star 2} + \eta^2 C_1^2 n)$$

$$\le \widehat{F}(w^\star) + \frac{\rho^{\star 2}}{\eta \sqrt{n}} + \frac{\rho^{\star 2} + \eta^2 C_1^2 n}{n \sqrt{n}}$$

In particular, in the above one can choose $w^\star := \arg\min_{w \in \mathbb{R}^p} F(w)$. Moreover, from the generalization guarantees,

$$\mathbb{E}_{\mathcal{S}}[F(w_T)] \le \mathbb{E}_{\mathcal{S}}[\widehat{F}(w_T)] + \frac{4\eta C_1^2}{n} \sum_{t=1}^{T-1} \mathbb{E}_{\mathcal{S}}[\widehat{F}(w_t)]$$

$$\le \left(1 + \frac{4\eta C_1^2}{\sqrt{n}}\right) \left(\mathbb{E}_{\mathcal{S}}[\widehat{F}(w^\star)] + \frac{\rho^{\star 2}}{\eta \sqrt{n}} + \frac{\rho^{\star 2} + \eta^2 C_1^2 n}{n \sqrt{n}}\right)$$

$$= \left(1 + \frac{4\eta C_1^2}{\sqrt{n}}\right) \left(F(w^\star) + \frac{\rho^{\star 2}}{\eta \sqrt{n}} + \frac{\rho^{\star 2} + \eta^2 C_1^2 n}{n \sqrt{n}}\right).$$

Note that $\eta \le \frac{1}{C_1^2 + C_2}$ by the descent lemma condition (Lemma 3). Therefore $\eta \le 1/C_1^2$. Without loss of generality we can assume $C_1 \ge 1$. This simplifies the above inequality as follows:

$$\mathbb{E}_{\mathcal{S}}[F(w_T)] \le \left(1 + \frac{4}{\sqrt{n}}\right) \left(F(w^\star) + \frac{\rho^{\star 2}}{\eta \sqrt{n}} + \frac{\rho^{\star 2}}{n \sqrt{n}} + \frac{1}{\sqrt{n}}\right)$$

$$= F(w^\star) + O\left(\frac{F(w^\star) + \rho^{\star 2}}{\sqrt{n}}\right).$$

This completes the proof of Theorem 2.3.

## E    PROOF OF THEOREM 2.4

We recall our setup and notation for Theorem 2.4. We consider in this section the one-hidden layer network with quadratic activation where $x \in \mathbb{R}^d, w_i \in \mathbb{R}^d$:

$$\Phi(w, x) = \frac{1}{2m} \sum_{i=1}^{m} a_i (x^\top w_i)^2$$

in the above, $a_i = \pm 1$ are fixed and half of the neurons have $a_i = 1$. For initialization of first layer weights $w_i \stackrel{\text{iid}}{\sim} N(0, \frac{I_d}{d})$ that is each $w_{ij} \stackrel{\text{iid}}{\sim} N(0, \frac{1}{d})$. The data points $x$ are uniformly drawn from the Rademacher distribution (of size $2^d$) that is $x(k) = \pm 1$ i.i.d and w.p. $1/2$ and the labels $y = x(1) \cdot x(2)$. Consider the linear loss where $f(u) = -u$ and $\widehat{F}(w) := \frac{1}{n} \sum_{j=1}^{n} f(y_j \Phi(x_j, w))$.

Consider stochastic gradient descent with the update rule as follows for each $w_i$:

$$\begin{aligned} w_i^{t+1} &= w_i^t - \eta \nabla \widehat{F}_i(w^t) \\ &= w_i^t - \frac{\eta a_i}{nm} \sum_{j=1}^{n} (x_j^\top w_i^t) x_j y_j f'(y_j \Phi(x_j, w)) \\ &= w_i^t + \frac{\eta a_i}{nm} \sum_{j=1}^{n} (x_j^\top w_i^t) x_j y_j \end{aligned}$$

Note that in the above the superscripts denote the iteration number and the subscript $i$ indicates the vector $w_i \in \mathbb{R}^d$ is associated with neuron $i$. Fixing the initialization, define $\mathbb{E}_x[w_i^t]$ as the outcome of GD after $t$ iterations on *population gradient*. Formally, $\mathbb{E}_x[w_i^t]$ is defined recursively as follows:

$$\mathbb{E}_x[w_i^{t+1}] := \mathbb{E}_x[w_i^t] - \eta \nabla F_i(\mathbb{E}_x[w^t]),$$

where $F(w) := \mathbb{E}_x[\widehat{F}(w)]$ is the test loss and define $\mathbb{E}_x[w^0] := w^0$. Based on previous derivations the update rule for the expected weights $\mathbb{E}_x[w_i^t]$ obeys the following:

$$\mathbb{E}_x[w_i^{t+1}] = \mathbb{E}_x[w_i^t] + \frac{\eta}{m} a_i \mathbb{E}_x \left[ \langle x, \mathbb{E}_x[w_i^t] \rangle xy \right].$$

for $t = 0$ :

$$\mathbb{E}_x[w_i^1] = w_i^0 + \frac{\eta}{m} a_i \mathbb{E}_x \left[ \langle x, w_i^0 \rangle xy \right].$$

For $t = 1$ :

$$\mathbb{E}_x[w_i^2] = \mathbb{E}_x[w_i^1] + \frac{\eta}{m} a_i \mathbb{E}_x \left[ \langle x, \mathbb{E}_x[w_i^1] \rangle xy \right].$$

and similarly $\mathbb{E}_x[w_i^t]$ is defined for all iterations $t$.

### E.1    CALCULATING $\mathbb{E}_x[w_i^1]$

For the expected gradient $g_i \in \mathbb{R}^d$ we have by noting $y = x(1)x(2)$:

$$g_i = -\mathbb{E}_x[\langle x, w_i^0 \rangle \cdot x \cdot x(1)x(2)]$$

for $g_i^3$ (we drop $i$ and $0$ for simplicity and denote the $k'$th element of $W_i^0$ with $W^k$ ):

$$\begin{aligned} -g_i^3 &= \mathbb{E}_x[(x(3)w^3 + \sum_{k \neq 3} x(k)w^k) \cdot x(3) \cdot x(1)x(2)] \\ &= \mathbb{E}_x[w^3 \cdot x(1)x(2) + x(3)x(1)x(2) \sum_{k \neq 3} x(k)w^k] \\ &= \mathbb{E}_x[w^3 \cdot x(1)x(2)] + \mathbb{E}_x[x(3)x(1)x(2) \sum_{k \neq 3} x(k)w^k] \\ &= 0 + \mathbb{E}_x[x(3) \sum_{k \neq 3} x(k)w^k] \, \mathbb{E}_x[x(1)x(2)] \\ &= 0. \end{aligned}$$

Therefore for any $k \neq 1, 2$ we have $g_i^k = 0$.

for $k = 1$ :

$$
\begin{aligned}
-g_i^1 &= \mathbb{E}_x[\langle x, w\rangle \cdot x(1) \cdot x(1)x(2)] \\
&= \mathbb{E}_x[w^1 x(1)x(2) + x(1)x(2)x(1)\sum_{k\neq 1} w^k x(k)] \\
&= \mathbb{E}_x[w^1 x(1)x(2)] + \mathbb{E}_x[x(1)x(2)x(1)w^2 x(2)] + \mathbb{E}_x[x(1)x(2)x(1)\sum_{k\neq 1,2} w^k x(k)] \\
&= w^1 \mathbb{E}_x[x(1)x(2)] + w^2 \mathbb{E}_x[x(1)x(2)x(1)x(2)] + \mathbb{E}_x[x(1)x(2)x(1)] \cdot \mathbb{E}_x[\sum_{k\neq 1,2} w^k x(k)] \\
&= w^2 \mathbb{E}_x[(x(1)x(2))^2] \\
&= w^2.
\end{aligned}
$$

Similarly, we derive that

$$
-g_i^2 = w_i^0(1)
$$

Thus, the expected weight after one step is

$$
\begin{aligned}
\mathbb{E}_x[w_i^1] &= w_i^0 + \frac{\eta a_i}{m}[w_i^0(2), w_i^0(1), 0, \cdots, 0] \\
&= \left[ w_i^0(1) + \frac{\eta a_i}{m} w_i^0(2), w_i^0(2) + \frac{\eta a_i}{m} w_i^0(1), w_i^0(3), \cdots, w_i^0(d) \right]
\end{aligned}
$$

### E.2 CALCULATING $\mathbb{E}_x[w_i^2]$

For $k = 1$ :

$$
\begin{aligned}
\mathbb{E}_x[w_i^2(1)] &= \mathbb{E}_x[w_i^1(1)] + \frac{\eta a_i}{m}\mathbb{E}_x[\langle x, \mathbb{E}_x[w_i^1]\rangle x(2)] \\
&= \mathbb{E}_x[w_i^1(1)] + \frac{\eta a_i}{m}\mathbb{E}_x[w_i^1(2)] \\
&= w_i^0(1) + \frac{\eta a_i}{m} w_i^0(2) + \frac{\eta a_i}{m}\left(w_i^0(2) + \frac{\eta a_i}{m} w_i^0(1)\right)
\end{aligned}
$$

Similarly we obtain for $k = 2$ :

$$
\begin{aligned}
\mathbb{E}_x[w_i^2(2)] &= \mathbb{E}_x[w_i^1(2)] + \frac{\eta a_i}{m}\mathbb{E}_x[\langle x, \mathbb{E}_x[w_i^1]\rangle x(1)] \\
&= \mathbb{E}_x[w_i^1(2)] + \frac{\eta a_i}{m}\mathbb{E}_x[w_i^1(1)] \\
&= w_i^0(2) + \frac{\eta a_i}{m} w_i^0(1) + \frac{\eta a_i}{m}\left(w_i^0(1) + \frac{\eta a_i}{m} w_i^0(2)\right)
\end{aligned}
$$

for $k \neq 1, 2$:

$$
\begin{aligned}
\mathbb{E}_x[w_i^2(k)] &= \mathbb{E}_x[w_i^1(k)] + \frac{\eta a_i}{m}\mathbb{E}_x[\langle x, \mathbb{E}_x[w_i^1]\rangle x(k)x(1)x(2)] \\
&= \mathbb{E}_x[w_i^1(k)]
\end{aligned}
$$

### E.3 CALCULATING $\mathbb{E}_x[w_i^t]$

In general, we observe the following pattern for arbitrary $t$: for $k \neq 1, 2$ :

$$
\mathbb{E}_x[w_i^{t+1}(k)] = \mathbb{E}_x[w_i^t(k)]
$$

and for $k = 1, 2$:

$$
\mathbb{E}_x[w_i^{t+1}(1)] = \mathbb{E}_x[w_i^t(1)] + \frac{\eta a_i}{m}\mathbb{E}_x[w_i^t(2)]
$$

$$
\mathbb{E}_x[w_i^{t+1}(2)] = \mathbb{E}_x[w_i^t(2)] + \frac{\eta a_i}{m}\mathbb{E}_x[w_i^t(1)]
$$

Therefore for general $t$, defining $\gamma = \eta a_i/m$, $\alpha(t) := \mathbb{E}_x[w_i^t(1)]$ and $\beta(t) := \mathbb{E}_x[w_i^t(2)]$ we have the following equations:

$$\alpha(t+1) = \alpha(t) + \gamma\beta(t)$$
$$\beta(t+1) = \beta(t) + \gamma\alpha(t).$$

It can be verified that the solution to the above equations take the following form:

$$\alpha(t) = \left( \sum_{r \in [t], r:even} \binom{t}{r} \gamma^r \right) \alpha(0) + \left( \sum_{r \in [t], r:odd} \binom{t}{r} \gamma^r \right) \beta(0)$$

$$\beta(t) = \left( \sum_{r \in [t], r:even} \binom{t}{r} \gamma^r \right) \beta(0) + \left( \sum_{r \in [t], r:odd} \binom{t}{r} \gamma^r \right) \alpha(0)$$

Therefore with replacement:

$$\mathbb{E}_x[w_i^t(1)] = \left( \sum_{r \in [t], r:even} \binom{t}{r} (\frac{\eta a_i}{m})^r \right) w_i^0(1) + \left( \sum_{r \in [t], r:odd} \binom{t}{r} (\frac{\eta a_i}{m})^r \right) w_i^0(2)$$

$$\mathbb{E}_x[w_i^t(2)] = \left( \sum_{r \in [t], r:even} \binom{t}{r} (\frac{\eta a_i}{m})^r \right) w_i^0(2) + \left( \sum_{r \in [t], r:odd} \binom{t}{r} (\frac{\eta a_i}{m})^r \right) w_i^0(1)$$

### E.4 EMPIRICAL GRADIENT FOR MULTIPLE GD STEPS

The results above hold when GD is applied on the entire data distribution. Let us consider the case of SGD on $n$ data points (on the loss $F(\cdot)$) for each iteration and bound the resulting noise.

Recall that

$$w^1 = w^0 - \eta\nabla F(w^0),$$
$$\mathbb{E}_x[w^1] = w^0 - \eta\mathbb{E}_x[\nabla F(w^0)],$$
$$w^1 = \mathbb{E}_x[w^1] - \eta(\nabla F(w^0) - \mathbb{E}_x[\nabla F(w^0)]).$$

Moreover, by how we defined $\mathbb{E}_x[w^t]$ for $t \geq 2$, we have

$$w^{t+1} = w^t - \eta\nabla F(w^t),$$
$$\mathbb{E}_x[w^{t+1}] = \mathbb{E}_x[w^t] - \eta\mathbb{E}_x[\nabla F(\mathbb{E}_x[w^t])],$$
$$w^{t+1} = \mathbb{E}_x[w^t] + (w^t - \mathbb{E}_x[w^t]) - \eta(\nabla F(w^t) - \mathbb{E}_x[\nabla F(\mathbb{E}_x[w^t])]) - \eta\mathbb{E}_x[\nabla F(\mathbb{E}_x[w^t])].$$

Overall, from the last two equations we have:

$$w^{t+1} - \mathbb{E}_x[w^{t+1}] = -\eta \sum_{\tau=0}^{t} \nabla F(w^\tau) - \mathbb{E}_x\left[\nabla F(\mathbb{E}_x[w^\tau])\right]$$

$$= -\eta \sum_{\tau=0}^{t} \underbrace{\nabla F(w^\tau) - \nabla F(\mathbb{E}_x[w^\tau])}_{\text{Term I}} + \underbrace{\nabla F(\mathbb{E}_x[w^\tau]) - \mathbb{E}_x[\nabla F(\mathbb{E}_x[w^\tau])]}_{\text{Term II}}.$$

We bound each term separately.

### E.5 TERM II:

Starting with the second term, recall that:

$$-\nabla F_i(\mathbb{E}_x[w^t]) = \frac{a_i}{nm} \sum_{j=1}^{n} \langle x_j, \mathbb{E}_x[w_i^t]\rangle x_j y_j$$

**Case 1:** For $k \neq 1, 2$ :

$$-\nabla F_i^k(\mathbb{E}_x[w^t]) = \frac{a_i}{nm} \sum_{j=1}^n \langle x_j, \mathbb{E}_x[w_i^t] \rangle x_j(k) x_j(1) x_j(2)$$

$$= \frac{a_i}{nm} \left\langle \sum_{j=1}^n x_j \cdot x_j(k) \cdot x_j(1) \cdot x_j(2), \mathbb{E}_x[w_i^t] \right\rangle$$

Denoting $v_j := x_j \cdot x_j(k) \cdot x_j(1) \cdot x_j(2)$ note that $v_j(r)$ are mutually independent for all $r \in d$ and $j \in [n]$. Moreover each $v_j(r)$ is distributed according to a Rademacher distribution and $Var(v_j(r)) = 1$.

Therefore, recalling the expression for $\mathbb{E}_x[w_i^t] = [\alpha_2^t w_i^0(1) + \alpha_1^t w_i^0(2), \alpha_1^t w_i^0(1) + \alpha_2^t w_i^0(2), w_i^0(3), \cdots, w_i^0(d)]$ where $\alpha_2^t := \sum_{r \in [t], r:even} \binom{t}{r} (\frac{\eta a_i}{m})^r$ and $\alpha_1^t := \sum_{r \in [t], r:odd} \binom{t}{r} (\frac{\eta a_i}{m})^r$ by expanding the summation above:

$$-\nabla F_i^k(\mathbb{E}_x[w^t]) = \frac{a_i}{nm} \Big( (\alpha_2^t \sum_j v_j(1) + \alpha_1^t \sum_j v_j(2)) w_i^0(1)$$

$$+ (\alpha_1^t \sum_j v_j(1) + \alpha_2^t \sum_j v_j(2)) w_i^0(2) + \sum_{k \neq 1,2} \sum_j v_j(k) w_i^0(k) \Big).$$

We use Hoeffding concentration inequality for bounding $\sum_j v_j(k) = O(\sqrt{n} \log(d))$ uniformly for every $k \neq 1, 2$. To be more concrete, by a union bound and Hoeffding it follows that uniformly over all $k \leq d$ and $t \leq T$:

$$\Pr\left( \sum_j v_j(k) \leq \sqrt{n} \log(dT) \right) \geq 1 - \exp\left( \log(dT) - 2\log^2(dT) \right) =: 1 - p_x.$$

Moreover, recall that $w_i^0(k) \sim N(0, 1/d)$ and :

$$\Pr\left( w_i^0(k) \leq \frac{\log(dm)}{\sqrt{d}} \right) \geq 1 - \frac{\exp(-\frac{1}{2}\log^2(dm))}{\sqrt{2\pi}\log(dm)}$$

By union bound we have for all $i \leq m, k \leq d$:

$$\Pr\left( w_i^0(k) \leq \frac{\log(dm)}{\sqrt{d}} \right) \geq 1 - \frac{\exp(-\frac{1}{2}\log^2(dm) + \log(dm))}{\sqrt{2\pi}\log(dm)} =: 1 - p_w.$$

Therefore w.p. $1 - p_x - p_w$ we have for all $t \leq T, i \leq m, k \leq d$: $\sum_j v_j(k) w_i^0(k) \leq \frac{\log(dm)\log(dT)\sqrt{n}}{\sqrt{d}}$
By Hoeffding w.p. $1 - p_x - p_w$ over data sampling and initialization:

$$\Pr\left( \sum_k \sum_j v_j(k) w_i^0(k) \leq \log(dm)\log(dT)\log(d)\sqrt{n} \right) \geq 1 - \exp(-\log^2(d)) =: 1 - p_1.$$

Similarly, we obtain $\Pr\left( \sum_j v_j(1) w_i^0(1) \leq \frac{\sqrt{n}}{\sqrt{d}}\log(dT)\log(dm) \right) \geq 1 - p_x - p_w$ Overall, the above calculations show that for some constant $C$ with probability $1 - C(p_1 - p_x - p_w)$ over initialization and data sampling it holds uniformly over all $k, i, t$:

$$|\nabla F_i^k(\mathbb{E}_x[w^t])| \lesssim \frac{\log(dT)\log(dm)}{nm} \left( \frac{2\sqrt{n}(|\alpha_1^t| + \alpha_2^t)}{\sqrt{d}} + \sqrt{n}\log(d) \right)$$

$$= \log(dT)\log(dm) \frac{|\alpha_1^t| + \alpha_2^t + \sqrt{d}\log(d)}{m\sqrt{nd}}.$$

**Case 2:**   Now assume $k = 1$ then in a similar way as above

$$-\nabla F_i^1(\mathbb{E}_x[w^t]) = \frac{a_i}{nm}\sum_{j=1}^n \langle x_j, \mathbb{E}_x[w_i^t]\rangle x_j(2)$$

$$= \frac{a_i}{m}\mathbb{E}_x[w_i^t(2)] + \frac{a_i}{nm}\sum_{j=1}^n\sum_{j'\neq 2}^d \mathbb{E}_x[w_i^t(j')]x_j(j')x_j(2)$$

$$= \frac{a_i}{m}\mathbb{E}_x[w_i^t(2)] + O(\log(dm)\log(dT)\frac{|\alpha_1^t|+\alpha_2^t + \sqrt{d}\log(d)}{m\sqrt{nd}}).$$

for $k = 2$ :

$$-\nabla F_i^2(\mathbb{E}_x[w^t]) = \frac{a_i}{m}\mathbb{E}_x[w_i^t(1)] + O(\log(dm)\log(dT)\frac{|\alpha_1^t|+\alpha_2^t + \sqrt{d}\log(d)}{m\sqrt{nd}}). \tag{50}$$

Therefore for the weights entering the $i$'th hidden neuron and for all $i \leq m, t \leq T$, the following concentration bound holds w.p. $1 - C(p_1 - p_x - p_w)$:

$$\left\|\nabla F_i(\mathbb{E}_x[w^t]) - \mathbb{E}_x[\nabla F_i(\mathbb{E}_x[w^t])]\right\| \lesssim \log(dm)\log(dT)\frac{|\alpha_1^t|+\alpha_2^t + \sqrt{d}\log(d)}{m\sqrt{n}}.$$

Therefore, as a result of Term II calculations, we deduce that with probability $1 - C(p_1 - p_x - p_w)$,

$$\forall i, t : \left\|\nabla F_i(\mathbb{E}_x[w^t]) - \mathbb{E}_x[\nabla F_i(\mathbb{E}_x[w^t])]\right\| \lesssim \log(dm)\log(dT)\frac{|\alpha_1^t|+\alpha_2^t + \sqrt{d}\log(d)}{m\sqrt{n}},$$

where $x^t$ denotes the data chosen at iteration $t$ of SGD and recall $w^0$ denotes the initialization weights.

### E.6   TERM I:

**Case 1:**   We first consider $k \neq 1, 2$. By applying Hoeffding's concentration inequality we obtain $\sum_{j=1}^n x_j(\ell)x_j(k)x_j(1)x_j(2) < \sqrt{n}\log(dT)$ uniformly for every $\ell \in [d], t \leq T$ w.p. $1 - \exp(\log(dT) - 2\log^2(dT)) =: 1 - p_x$. Applying Hoeffding again yields,

$$\nabla F_i^k(w^t) - \nabla F_i^k(\mathbb{E}_x[w^t]) = \frac{a_i}{nm}\sum_{j=1}^n \langle x_j, \mathbb{E}_x[w_i^t] - w_i^t\rangle x_j(k)x_j(1)x_j(2)$$

$$= \frac{a_i}{nm}\left\langle \sum_{j=1}^n x_jx_j(k)x_j(1)x_j(2), \mathbb{E}_x[w_i^t] - w_i^t\right\rangle$$

$$\leq \frac{\|\mathbb{E}_x[w_i^t] - w_i^t\|}{\sqrt{n}m}\log(dT)\log(dmT),$$

w.p. $1 - \exp(\log(dmT) - 2\log^2(dmT)) - \exp(\log(dT) - 2\log^2(dT)) =: 1 - p_2 - p_x$ uniformly over all $k \leq d, i \leq m, t \leq T$ where in the above we assumed $\mathbb{E}_x[w_i^t] - w_i^t$ as a fixed vector.

Denote $v^t := \mathbb{E}_x[w_i^t] - w_i^t$. Concretely, we have the following bound for $k \neq 1, 2$ on Term I:

$$\Pr_{x^t}\left(\forall i, k, t : \nabla F_i^k(w^t) - \nabla F_i^k(\mathbb{E}_x[w^t]) \leq \frac{\|v^t\|}{\sqrt{n}m}\log(dT)\log(dmT)\Big|v^t\right) \geq 1 - p_2 - p_x.$$

Therefore,

$$\Pr_{x^t, v^t}\left(\forall i, k, t : \nabla F_i^k(w^t) - \nabla F_i^k(\mathbb{E}_x[w^t]) \leq \frac{\|v^t\|}{\sqrt{n}m}\log(dT)\log(dmT)\right) \geq 1 - p_2 - p_x.$$

**Case 2:** For $k = 1$ :

$$\nabla F_i^k(w^t) - \nabla F_i^k(\mathbb{E}_x[w^t]) = \frac{a_i}{nm} \sum_{j=1}^n \langle x_j, \mathbb{E}_x[w_i^t] - w_i^t \rangle x_j(2)$$

$$= \frac{a_i}{nm} \langle \sum_{j=1}^n x_j x_j(2), \mathbb{E}_x[w_i^t] - w_i^t \rangle$$

$$= \frac{a_i}{m} v^t(2) + \frac{a_i}{m\sqrt{n}} \|v^t(\ell \neq 2)\| \cdot \log(dT) \log(mT),$$

w.p. $1 - p_2 - p_x$.

Similarly for $k = 2$ with the same probability it holds,

$$\nabla F_i^k(w^t) - \nabla F_i^k(\mathbb{E}_x[w^t]) = \frac{a_i}{nm} \sum_{j=1}^n \langle x_j, \mathbb{E}_x[w_i^t] - w_i^t \rangle x_j(1)$$

$$= \frac{a_i}{nm} \langle \sum_{j=1}^n x_j x_j(1), \mathbb{E}_x[w_i^t] - w_i^t \rangle$$

$$= \frac{a_i}{m} v^t(1) + \frac{a_i}{m\sqrt{n}} \|v^t(\ell \neq 1)\| \cdot \log(dT) \log(mT)$$

### E.7 COMBINING TWO TERMS

Recall we defined $v^t = \mathbb{E}_x[w_i^t] - w_i^t$ where $v^t \in \mathbb{R}^d$. Denote its entries by $v^t(k)$ for $k \leq d$. Define $v^0 = 0$. Then, from the last two sections we obtain the following system of equations which hold w.p. $1 - C(p_1 - p_x - p_2 - p_w)$ for all $t \geq 1$:

$$|v^t(1)| \leq \eta \sum_{\tau=0}^{t-1} \frac{|v^\tau(2)|}{m} + \frac{\|v^\tau(\ell \neq 2)\| \log(dT) \log(mT)}{m\sqrt{n}} + \frac{|\alpha_1^\tau| + \alpha_2^\tau + \sqrt{d} \log(d)}{m\sqrt{nd}} \log(dm) \log(dT)$$

$$|v^t(2)| \leq \eta \sum_{\tau=0}^{t-1} \frac{|v^\tau(1)|}{m} + \frac{\|v^\tau(\ell \neq 1)\| \log(dT) \log(mT)}{m\sqrt{n}} + \frac{|\alpha_1^\tau| + \alpha_2^\tau + \sqrt{d} \log(d)}{m\sqrt{nd}} \log(dm) \log(dT)$$

$$|v^t(k)| \leq \eta \sum_{\tau=0}^{t-1} \frac{\|v^\tau\| \log(dT) \log(dmT)}{m\sqrt{n}} + \frac{|\alpha_1^\tau| + \alpha_2^\tau + \sqrt{d} \log(d)}{m\sqrt{nd}} \log(dm) \log(dT), \quad \forall k \neq 1, 2$$

Note that $|\alpha_1^\tau| + \alpha_2^\tau = 2^\tau$. Simplify the above by assuming $\eta = m$ and denoting $\gamma_\tau := \frac{2^\tau + \sqrt{d} \log(d)}{\sqrt{nd}} \log(dm) \log(dT)$. Then the equations above simplify to:

$$|v^t(1)| \leq \sum_{\tau=0}^{t-1} |v^\tau(2)| + \frac{\|v^\tau\| \log^2(dmT)}{\sqrt{n}} + \sum_{\tau=0}^{t-1} \gamma_\tau$$

$$|v^t(2)| \leq \sum_{\tau=0}^{t-1} |v^\tau(1)| + \frac{\|v^\tau\| \log^2(dmT)}{\sqrt{n}} + \sum_{\tau=0}^{t-1} \gamma_\tau$$

$$|v^t(k)| \leq \sum_{\tau=0}^{t-1} \frac{\|v^\tau\| \log^2(dmT)}{\sqrt{n}} + \sum_{\tau=0}^{t-1} \gamma_\tau, \quad \forall k \neq 1, 2$$

Define $z^t := \max_k |v^t(k)|$. Then since the RHS in the third equations is smaller than the two first equations we deduce that:

$$z^t \leq \sum_{\tau=0}^{t-1} z^\tau + \frac{z^\tau \sqrt{d} \log^2(dmT)}{\sqrt{n}} + \frac{2^\tau + \sqrt{d} \log(d)}{\sqrt{nd}} \log^2(dmT) \tag{51}$$

To proceed in simplifying the equation above, we need the following lemma which can be straight-forwardly proved by induction.

**Lemma 6.** *Let* $v_k, \beta, \gamma_k \in \mathbb{R}$. *If* $v_0 = 0$ *and for every* $t < T$:

$$v_{t+1} \leq \beta \sum_{k=0}^{t} v_k + \sum_{k=0}^{t} \gamma_k \tag{52}$$

*Then, it can be checked that this results in:*

$$v_T \leq \sum_{k=0}^{T-1} (\beta + 1)^k \gamma_{T-k-1}. \tag{53}$$

Therefore, continuing from Eq. 51:

$$z^t \leq \sum_{\tau=0}^{t-1} (2 + \frac{\sqrt{d} \log^2(dmT)}{\sqrt{n}})^\tau \cdot \frac{2^{t-\tau-1} + \sqrt{d} \log(d)}{\sqrt{nd}} \log^2(dmT)$$

$$\leq \frac{t \log^2(dmT)}{\sqrt{nd}} (2 + \frac{\sqrt{d} \log^2(dmT)}{\sqrt{n}})^{t-1} + \frac{(2 + \frac{\sqrt{d} \log^2(dmT)}{\sqrt{n}})^t \log^3(dmT)}{\sqrt{n}(1 + \frac{\sqrt{d} \log^2(dmT)}{\sqrt{n}})}$$

Assume $t \leq \log(d)$ and $n \geq d \cdot \log^{2c}(d)$ for some constant $c$. Then,

$$z^t \leq \frac{d}{d \log^{c-3}(d)} + \frac{2d}{\sqrt{d} \log^{c-3}(d)}$$

$$\leq \frac{3\sqrt{d}}{\log^{c-3}(d)}$$

where in the above we used the fact that for $t \leq T \leq \log(d)$ and any $m = O(\text{poly}(d))$:

$$(2 + \frac{\sqrt{d} \log^2(dmT)}{\sqrt{n}})^{t-1} = (2 + \frac{1}{\log^{c-2}(d)})^{t-1} \leq (2 + \frac{1}{\log(d)})^{t-1} \leq d$$

Therefore

$$\max_k |v^t(k)| = O(\frac{\sqrt{d}}{\log^{c-3}(d)})$$

We next show that $\|v^t(3:d)\|$ is much smaller. To see this, we replace our derived bound in the third equation above. define $\beta^t := \|v^t(3:d)\|$ and note that $\beta^t \leq \sqrt{d}|v^t(k)|$ for some $k \in [3, d]$. Thus:

$$\beta^t \leq \sum_{\tau=0}^{t-1} \frac{\|v^\tau\| \log^2(dmT)\sqrt{d}}{\sqrt{n}} + \sqrt{d} \sum_{\tau=0}^{t-1} \gamma_\tau$$

$$\leq \sum_{\tau=0}^{t-1} \frac{\|v^\tau\|}{\log^{c-2}(d)} + \sqrt{d} \sum_{\tau=0}^{t-1} \gamma_\tau$$

$$\leq \sum_{\tau=0}^{t-1} \frac{\sqrt{(\beta^\tau)^2 + \frac{d}{\log^{2c-6}(d)}}}{\log^{c-2}(d)} + \sqrt{d} \sum_{\tau=0}^{t-1} \gamma_\tau$$

$$\leq \sum_{\tau=0}^{t-1} \frac{\beta^\tau + \frac{\sqrt{d}}{\log^{c-3}(d)}}{\log^{c-2}(d)} + \sum_{\tau=0}^{t-1} \frac{2^\tau + \sqrt{d} \log(d)}{\sqrt{n}} \log^2(dmT)$$

Using the lemma for the recursive summation (Lemma 6) again:

$$\beta^t \leq \sum_{\tau=0}^{t-1} (1 + \frac{1}{\log^{c-2}(d)})^\tau (\frac{\sqrt{d}}{\log^{2c-5}(d)} + \frac{2^{t-\tau-1} + \sqrt{d} \log(d)}{\sqrt{n}} \log^2(dmT))$$

$$\leq (1 + \frac{1}{\log^{c-2}(d)})^t \frac{\sqrt{d}}{\log^{c-3}(d)} + \frac{2^{t-1} t}{\sqrt{d} \log^{c-2}(d)} + (1 + \frac{1}{\log^{c-2}(d)})^t \log^2(dmT)$$

where in the above we used the conditions on $m, T$. By using them again and recalling $t \leq T \leq \log(d)$, the above simplifies into:

$$\beta^t \leq \frac{2\sqrt{d}}{\log^{c-3}(d)} + \frac{\sqrt{d}}{\log^{c-3}(d)} + 2\log^2(dmT)$$

$$= O(\frac{\sqrt{d}}{\log^{c-3}(d)})$$

recall that $\mathbb{E}_x[w_i^t]$ had the following form

$$\mathbb{E}_x[w_i^t(1)] = \left( \sum_{r \in [t], r:even} \binom{t}{r} (\frac{\eta a_i}{m})^r \right) w_i^0(1) + \left( \sum_{r \in [t], r:odd} \binom{t}{r} (\frac{\eta a_i}{m})^r \right) w_i^0(2)$$

$$= \frac{1}{2}((1+a_i)^t + (1-a_i)^t) w_i^0(1) + \frac{1}{2}((1+a_i)^t - (1-a_i)^t) w_i^0(2)$$

$$\mathbb{E}_x[w_i^t(2)] = \left( \sum_{r \in [t], r:even} \binom{t}{r} (\frac{\eta a_i}{m})^r \right) w_i^0(2) + \left( \sum_{r \in [t], r:odd} \binom{t}{r} (\frac{\eta a_i}{m})^r \right) w_i^0(1)$$

$$= \frac{1}{2}((1+a_i)^t + (1-a_i)^t) w_i^0(2) + \frac{1}{2}((1+a_i)^t - (1-a_i)^t) w_i^0(1)$$

$$\mathbb{E}_x[w_i^t(k)] = w_i^0(k), k > 2.$$

where the last line is derived by choosing w.l.o.g. $\eta = m$ :

$$\mathbb{E}_x[w_i^t(1)] = 2^{t-1}(w_i^0(1) + a_i w_i^0(2)),$$
$$\mathbb{E}_x[w_i^t(2)] = 2^{t-1}(w_i^0(2) + a_i w_i^0(1)),$$
$$\mathbb{E}_x[w_i^t(k)] = w_i^0(k), \ k > 2.$$

therefore,

$$|\mathbb{E}_x[w_i^t(1)]| = \Theta(\frac{2^t}{\sqrt{d}}),$$

$$|\mathbb{E}_x[w_i^t(2)]| = \Theta(\frac{2^t}{\sqrt{d}}),$$

$$|\mathbb{E}_x[w_i^t(k)]| = \Theta(\frac{1}{\sqrt{d}}).$$

Select $t = \log(d)$. Then, w.h.p. it holds that

$$|\mathbb{E}_x[w_i^t(1)]| = \Theta(\sqrt{d}),$$
$$|\mathbb{E}_x[w_i^t(2)]| = \Theta(\sqrt{d}),$$
$$|\mathbb{E}_x[w_i^t(k)]| = \Theta(\frac{1}{\sqrt{d}}).$$

The calculations above indicate that the signal strength is already larger than the noise after $\log(d)$ SGD steps. We are now ready to compute $y\Phi(w^t, x)$. Assume for simplicity and without loss of generality that $x(1) = x(2) = y = 1$ as other cases lead to the same result. Recall $v_i^t(k) := \mathbb{E}_x[w_i^t(k)] - w_i^t(k)$. Then,

$$y\Phi(w^t, x) = \frac{1}{m}\sum_{i=1}^{m} a_i \left( w_i^t(1) + w_i^t(2) + \sum_{k=3}^{d} w_i^t(k)x(k) \right)^2$$

$$= \frac{1}{m}\sum_{i=1}^{m} a_i \left( d(w_i^0(1) + w_i^0(2))(a_i + 1) - v_i^t(1) - v_i^t(2) + \sum_{k\geq 3}(w_i^0(k) - v_i^t(k))x(k) \right)^2$$

$$= \frac{1}{m}\sum_{i\in P} \left( 2d(w_i^0(1) + w_i^0(2)) - v_i^t(1) - v_i^t(2) + \sum_{k\geq 3}(w_i^0(k) - v_i^t(k))x(k) \right)^2$$

$$+ \frac{1}{m}\sum_{i\in N} \left( v_i^t(1) + v_i^t(2) + \sum_{k\geq 3}(v_i^t(k) - w_i^0(k))x(k) \right)^2,$$

where $P$ and $N$ denote the set of neurons for which $a_i = 1$ and $a_i = -1$, respectively.

Note that the last term in each summation can be simplified as follows:

$$\sum_{k\geq 3}(w_i^0(k) + v_i^t(k))x(k) = O(\|v_i^t\|) = O\left(\beta^t\right) = O\left(\frac{\sqrt{d}}{\log^{c-3}(d)}\right).$$

Concretely, by Hoeffding w.p. $1 - C(p_1 - p_2 - p_x - p_w)$ over data sampling and initialization :

$$\Pr_x \left( \sum_{k\geq 3}\left(w_i^0(k) + v_i^t(k)\right)x(k) > \frac{\sqrt{d}}{\log^{\frac{c-4}{2}}(d)} \right) \leq e^{-\log(d)} = \frac{1}{d}. \tag{54}$$

Define $\zeta_i^t := v_i^t(1) + v_i^t(2) + \sum_{k\geq 3}(v_i^0(k) - w_i^t(k))x(k)$. Then,

$$y\Phi(w^t, x) = \frac{1}{m}\sum_{i\in P}\left(2d(w_i^0(1) + w_i^0(2)) - \zeta_i^t\right)^2 + \frac{1}{m}\sum_{i\in N}\left(\zeta_i^t\right)^2$$

$$= \frac{1}{m}\sum_{i\in P} 4d^2(w_i^0(1) + w_i^0(2))^2 - 4d\zeta_i^t(w_i^0(1) + w_i^0(2)) + \frac{1}{m}\sum_{i\in[m]}(\zeta_i^t)^2$$

$$\geq \frac{1}{m}\sum_{i\in P} 4d^2(w_i^0(1) + w_i^0(2))^2 - 4d|\zeta_i^t|\cdot|w_i^0(1) + w_i^0(2)|. \tag{55}$$

Moreover, we had w.p. at least $1 - C(p_1 + p_2 + p_x + p_w)$ that $|v_i^t(1)| + |v_i^t(2)| < \frac{C\sqrt{d}}{\log^{c-3}(d)}$ for some absolute constant $C$. Adding these two together, we deduce that w.p. $1 - o_d(1)$ it holds uniformly over $i, t$,

$$|\zeta_i^t| < \frac{C\sqrt{d}}{\log^{\frac{c-4}{2}}(d)}.$$

It remains to obtain lower- and upper-bounds on $(w_i^0(1) + w_i^0(2))^2$ as required by Eq. 55. To proceed, note that $\frac{d}{2}(w_i^0(1) + w_i^0(2))^2 \sim \chi^2(1)$. Therefore with standard concentration bounds for $\chi^2(1)$ and $\chi^2(m)$ distributions (e.g., Laurent & Massart (2000)) we find for any $u \geq 0$,

$$\Pr\left(\frac{d}{2}(w_i^0(1) + w_i^0(2))^2 - 1 \geq 2\sqrt{u} + 2u\right) \leq e^{-u}$$

$$\Pr\left(m - \frac{d}{2}\sum_{i=1}^{m}(w_i^0(1) + w_i^0(2))^2 \geq 2\sqrt{mu}\right) \leq e^{-u}$$

By selecting $u = \log^2(d)$ in the first inequality above, we find w.p. at most $e^{\log(m)-\log^2(d)}$ it holds uniformly for all $i \leq m$:

$$(w_i^0(1) + w_i^0(2))^2 \geq \frac{2}{d} + \frac{4}{d}\sqrt{\log^2(d)} + \frac{4}{d}\log^2(d),$$

and by noting that $|P| = |N| = m/2$ and selecting $u = m/16$ in the second inequality we have w.p. at least $1 - e^{-\frac{m}{16}}$,

$$\frac{1}{m}\sum_{i \in P}(w_i^0(1) + w_i^0(2))^2 \geq (1 - \frac{1}{\sqrt{2}})\frac{1}{d}.$$

Hence w.p. $1 - e^{\log(m)-\log^2(d)} - e^{-\frac{m}{16}} - C(e^{-\log^2(d)} + e^{\log(dmT)-2\log^2(dmT)} + e^{\log(dT)-2\log^2(dT)} + e^{-\frac{1}{2}\log^2(dm)+\log(dm)})$ over data sampling and initialization,

$$\Pr_{x,y}\left(y\Phi(w^t, x) \geq d - \frac{4Cd}{\log^{\frac{c-6}{2}}(d)}\right) \geq 1 - 1/d. \tag{56}$$

For the theorem's choice of $T = \log(d), m = O(\text{poly}(d))$ it is deduced that $e^{\log(m)-\log^2(d)} + C(e^{-\log^2(d)} + e^{\log(dmT)-2\log^2(dmT)} + e^{\log(dT)-2\log^2(dT)} + e^{-\frac{1}{2}\log^2(dm)+\log(dm)}) = o_d(1)$. Moreover, for the theorem's choice $c = 7$ and large enough $d$ it holds that $d - 4Cd/\log^{\frac{c-6}{2}}(d) \geq 0$. This leads to the following lower bound on test accuracy,

$$\Pr_{x,y}\left(y\Phi(w^t, x) \geq 0\right) \geq 1 - \frac{1}{d}, \tag{57}$$

with probability at least $1 - e^{-\frac{m}{16}} - e^{\log(m)-\log^2(d)} - o_d(1)$ over data sampling and initialization. This completes the proof of Theorem 2.4.