# OpenReview forum: "Sharper Guarantees for Learning Neural Network Classifiers with Gradient Methods"
_ICLR.cc/2025/Conference — ICLR 2025 Poster_

### Official Review · Reviewer_zVMC · 2024-11-01

**Soundness:** 3
**Presentation:** 2
**Contribution:** 3
**Rating:** 6
**Confidence:** 3

**Summary:**

This paper studies convergence and generalization for deep neural networks with smooth activation trained by gradient descent. This paper shows several improved results on test error compared with previous works for the following scenarios: (1) noiseless data with separable NTK feature, and (2) noisy data and (3) XOR distribution.

**Strengths:**

This work presents several improved results from prior works. In particular, the authors are able to show an improved bound on the network width by requiring $\lvert w^\star - w_0 \rvert \lesssim m^{O(1/L)}$, which implies that the NTK regime can happen under a far more relaxed conditions than what previous work suggested. Such results can be important and interesting. The authors then utilize this improvement to derive improved bounds for several classical NTK settings like NTK separable data and noisy data.

**Weaknesses:**

Although the results in this work are impressive, a more detailed description on how the authors achieved this improvement is needed. In particular, the authors mentioned in line 84, 85 that the reason for the improvement in the network's width is by exploring the objective's Hessian structure in the gradient descent path. This sounds very interesting. However, I didn't find any high level discussion/description on this in the main text. I would like the authors to explain how they get this improvement. How does your analysis compare with the prior work that also utilized the Hessian structure? What is the novelty in your analysis such that you can achieve this improvement?

**Questions:**

None.

**Details Of Ethics Concerns:**

None.

---

> ### Author Response · Authors · 2024-11-19
>
> We thank the reviewer for their time and their valuable comments on our submission.
>
> The result in [Chen et al. 2021] assumes $\|w^*-w_0\|=O(1)$ which leads to the first-order approximation for the model i.e., $\Phi(w) = \Phi(w_0) + \nabla \Phi(w_0) ^\top (w-w_0)$ being accurate in the constant-radius ball around initialization. On the other hand, our analysis uses second order information and the Hessian’s spectral-norm bound to show that as long as $\|w^*-w_0\|=O(m^{O(1/L)})$, the objective has favorable quasi convex-like properties (see pages 16-17) which ultimately leads to global convergence and good generalization. Moreover, using smoothness of the activation enables us to derive generalization bounds based on algorithmic stability which leads to finer generalization bounds compared to Rademacher complexity based bounds in [Chen et al. 2021] (please also see lines 128-168). Approaches based on Hessian spectral norm of deep nets has been used for obtaining training convergence of GD for square loss (cf. Remark 2.2). In particular, here the self-bounded property of the minimum eigen-value of the Hessian, (i.e., that the lower bound on $\lambda_{min}(\nabla^2 \hat F(w))$ is proportional to $\hat F(w)$) enables us to obtain smaller width lower bounds of poly-logarithmic order compared to the least-squares case which requires polynomial width.

---

> > ### Comment · Reviewer_zVMC · 2024-11-26
> >
> > Thank you for your response. I have read the author's response along with other reviews. I will keep my score.

---

> > > ### Author Response · Authors · 2024-11-26
> > > **Thank you for your feedback**
> > >
> > > Thank you very much for your thoughtful review and for highlighting the contributions of our work. We will be happy to address any further questions you may have.

---

### Official Review · Reviewer_Udj5 · 2024-11-02

**Soundness:** 3
**Presentation:** 3
**Contribution:** 3
**Rating:** 6
**Confidence:** 3

**Summary:**

This paper studies the data-dependent convergence and generalization of gradient methods in neural networks with smooth activations. The authors provide a tighter, algorithm-dependent bound on excess risk for deep networks trained with logistic loss, improving on Rademacher complexity-based bounds. These results emphasize the role of initialization, particularly in small-width networks and margin-separable data.

**Strengths:**

Overall, the contribution of this paper is great filling the gap between practice and theory.

**Weaknesses:**

Please see the questions below

**Questions:**

1.	Table in page 2 is mis-labeled as Table 2.
2.	The authors claim that leveraging the Hessian structure in the gradient descent path is the key for their theoretical improvements over the past literature. But I feel the mathematical intuition is not well stated in the main paper. When it is compared with the previous literature (for instance Chen et al, 2021), how does considering the hessian structure benefit the author to prove Theorem 2.1?
3.	In least-squares regression, the Neural Tangent Kernel (NTK) benefits from the fact that gradient descent keeps weight updates close to their initialization, which linearizes the dynamics in parameter space and makes the analysis more tractable. I suspect this approach will similarly apply to classification tasks if the authors use the NTK framework. Given this, the role of initialization mentioned in the main paper isn't particularly surprising. Could the authors elaborate on the unique insights that arise when studying classification problems, which might not be evident when analyzing regression problems, specifically in terms of the role of initialization?
4.     Somehow, I felt Subsection 2.2 is disconnected to the remaining part of the paper, as the problem settings seem to be different from the previous parts. I suspect this is because of technical difficulties authors might have encountered, but it would be great if authors can make connections between this Subsection and previous parts. For example, how are the particular techniques from earlier sections linked to the XOR distribution analysis in 2.2?
5.	In Theorem 2.4, the step size is equal to the width of the network. Can it be arbitrarily large? Are there any restrictions on the step size?

**Details Of Ethics Concerns:**

This is a theoretical paper, therefore there are no ethical concerns regarding the contents of the paper.

---

> ### Author Response · Authors · 2024-11-19
>
> We thank the reviewer for their time and their valuable comments on our submission.
>
> 2-The result in (Chen et al. 2021) assumes $\|w^*-w_0\|=O(1)$ which leads to the first-order approximation for the model i.e., $\Phi(w) = \Phi(w_0) + \nabla \Phi(w_0) ^\top (w-w_0)$ being accurate in the constant-radius ball around initialization. On the other hand, our analysis uses the second order information and the fact that Hessian’s minimum eigen-value at any $w$ is larger than $-\frac{\|w-w_0\|^{3L}}{\sqrt{m}} \hat F(w)$ to show that as long as $\|w^*-w_0\|=O(m^{O(1/L)})$, the objective has favorable quasi convex-like properties along the GD path.  Moreover, using the Hessian structure of smooth networks enables obtaining the algorithmic-dependent  generalization-gap bounds based on algorithmic stability. On the other hand, (Chen et al. 2021) uses Rademacher-complexity based bounds for deep nets which lead to the undesirable terms in the generalization loss. Since for algorithmic stability the generalization is solely dependent on the training loss performance, the generalization loss has the same favorable properties. This ultimately leads to tighter bounds.
>
> 3- Similar to the regression tasks, for classification tasks the weights are also close to initialization in the NTK regime. As mentioned earlier, the guarantees hold as long as the distance between the target weights and initialization satisfies $\|w^*-w_0\| = O(m^{1/L})$. We note that the convergent analysis is different from regression tasks using least squares method. Here the objective function has approximately quasi-convex-like properties. In particular, the self-bounded property of the minimum eigen-value of the Hessian, (i.e., that the lower bound on $\lambda_{min}(\nabla^2 \hat F(w))$ is proportional to $\hat F(w)$) enables us to obtain smaller width lower bounds of poly-logarithmic order compared to the least-squares case which requires polynomial width.
>
> 4- The goal in section 2.2 was to discuss the benefits of escaping the NTK regime in obtaining smaller width, iteration and sample complexity requirements. Indeed, there is a difference in problem setups between the sections as the proof specifically holds for the XOR dataset and two-layer nets. Extensions of the XOR problem’s results to multi-layer networks or to more general datasets are interesting future directions.
>
>
>
> 5- The step-size can be chosen such that $\frac{\eta}{m \cdot\text{polylog}(d)} = o_d(1)$. However, for larger step sizes, the noise caused by SGD can dominate the signal (i.e., the expected weights), which makes our test bound invalid.

---

> > ### Comment · Reviewer_Udj5 · 2024-11-25
> > **Thanks for the response. Authors**
> >
> > Thanks for the responses. I will keep the score as it is.

---

> > > ### Author Response · Authors · 2024-11-26
> > > **Thanks for your feedback**
> > >
> > > Thank you for your thoughtful questions and comments. We would be happy to address any further questions or concerns you may have.

---

### Official Review · Reviewer_gGtd · 2024-11-02

**Soundness:** 3
**Presentation:** 2
**Contribution:** 3
**Rating:** 8
**Confidence:** 4

**Summary:**

This paper studies the training convergence guarantee and the generalization error of deep MLPs with quadratic activation when applied to binary classification tasks. The paper considers the parameterization of the neural network in the NTK regime, and shows a tigher generalization bound together with a smaller over-parameterization requirement compared with previous work. In addition to the analysis of the noiseless separable data scheme, the paper also considers a noisy data scheme and a XOR classification task. In particular, the paper shows that for the XOR classification, using a large step size and effectively escape the NTK regime and learns useful features. The paper also provided experimental results to verify its theoretical conclusion.

**Strengths:**

1. The paper obtains a tigher bound on the generalization error compared with previous work. In particular, while previous work usually achieves a $O(\frac{1}{\sqrt{n}})$ generalization error, this paper achieves a $O(\frac{1}{n})$ error.

2. In the noiseless data scheme, the paper derives favorable convergence guarantee under only a constant over-parameterization requirement.

3. The paper has a nice extension of their theoretical work from the noiseless data scheme to the noisy data scheme as well as the XOR classification task.

**Weaknesses:**

1. In Theorem 2.1, the over-parameterization requirement scales with $\\Omega(\\rho^{*6L+4})$. Later in the discussion, the paper stated $\\rho = O(\\log n / \\gamma)$, where $\\gamma$ is the margin width. When we treat $L$, the network depth, not as a constant, the over-parameterization requirement can be as large as $\\Omega(\\frac{(\\log n)^{6L+4}}{\\gamma^{6L+4}})$, which might be unfavorable.

2. The setting studied in the paper are not clearly explained, and some notations are not introduced before use. For instance, it is not clear what is $\mathcal{S}$ is in Eq.(5), and what it means to take expectation over it. Moreover, in Section 2.1.2, it is not clear how the noise in the data are generated.

3. In Assumption 1, the separability condition is defined over the tangent feature at initialization. Because the separability is defined in this random feature model, it would naturally imply an implicit lower bound on the over-parameterization: for instance, the more parameters we have, the more random features we will have, and a higher probability the data will be separable when mapped to the feature space.

4. Some theoretical results seems confusing; see Questions below.

**Questions:**

1. In Eq. (5), consider the special case of $n=1$, where we have only one sample. Suppose we initialize $w_0$ at $w^\star$ and suppose $w^\star$ achieves zero training error. Then by Eq. (5), the generalization error will be $0$. This means that even in the case where we have only one trainig sample, every neural network parameter that achieves zero training error will generalize well, which is counter-intuitive. Am I missing something here?

2. In Eq. (6), plugging in $\hat{F} \leq \frac{4\rho^{*2}}{\eta T}$, should we have an additional $\log T$ on the right-hand side due to a summation of $\frac{1}{t}$ for $t = 1,\dots, T-1$?

3. In the cited paper Lei and Ying (2020), I did not found Lemma B.3, which is used in the proof of the generalization error. Is the paper referring to a different lemma? Also, it seems that the leave-one-out error in Lei and Ying is defined by replacing the $i$th sample with another random sample, not deleting the $i$th sample.

---

> ### Author Response · Authors · 2024-11-19
>
> We thank the reviewer for their time and their valuable comments on our submission.
>
> 2-  Thanks for bringing up this issue. We will clarify in the revision. $\mathcal{S}$ denotes the training set. The generalization loss is a random variable depending on the choice of training set and the expectation is with respect to the randomness in selecting the training set.
> There is no condition on the noise generation for Section 2.1.2. Unlike the results in thm 2.1. which holds in the interpolation regime, here for thm 2.3. $F(w^*)$ (that is the optimal test error) can be non-vanishing as it is the case for noisy (nonseparable) data. In fact, the result holds for any such $F(w^*)$.
>
> 3- Indeed, assumption 1 holds with high-probability and a width lower bound is implicit for this assumption to hold. However, this assumption is standard in literature and used in several recent works in deep learning theory [Ji and Telgarsky 2020, Chen et al. 2021]. Alternatively, one could assume the infinite width NTK separability condition and derive the finite width NTK separability from that.
>
> Q1- The results for generalization loss in Thm 2.1 hold if the conditions of the theorem (that is Eq. 2 and Eq. 3) are true for ‘every n samples’ from the distribution. For the case in your question, with $w_0$ being fixed, this can only happen if $w^*$ is the minimizer of the loss on the entire distribution. We also note that this justifies the fact that the generalization loss in Eq. 5 holds in expectation over the randomness of the training set rather than for a fixed training set. We apologize for the confusion and we’ve now added a sentence in Thm 2.1. clarifying this.
>
> Q2- In fact, our training loss guarantees hold for the averaged training loss $\frac{1}{T} \sum F(w_t)$ (cf. Thm B.1 in the appendix) and the last iterate guarantee stems from the descent lemma as $F(w_T)\le F(w_t)$ for all $t\le T.$ Thus, the logarithmic term can be avoided by using the averaged training loss guarantee. Thanks for bringing this up; we have now clarified in the revision.
>
>
>
> Q3- We apologize for the confusion and have fixed the theorem reference in the revision (we are referring to Thm 2 in [Lei and Ying 2020]). The reviewer is correct that their notion of stability requires replacing the i’th sample rather than deleting it. However, the statement of our lemma 5 is correct and the proof is similar (e.g., see Lemma 6 in [1])
>
> [1] Schliserman, Matan, and Tomer Koren. "Stability vs implicit bias of gradient methods on separable data and beyond." Conference on Learning Theory. PMLR, 2022.

---

> > ### Comment · Reviewer_gGtd · 2024-11-26
> >
> > Thank you so much for your response and the corresponding changes in the new version of the paper. I believe that most of my concerns are resolved, leaving only the dependency of the overparameterization on the depth which might be unfavorable. However, I understand that this limitation persist in a lot of the theory works of neural networks, and, despite this disadvantage, I believe that this paper makes a good contribution to the field. Therefore, I believe that this paper deserves a clear accept.

---

> ### Author Response · Authors · 2024-11-26
> **Thank you for your feedback**
>
> Thank you very much for your insightful feedback, which has greatly contributed to improving the clarity and quality of our paper! We are happy that our response addressed most of your concerns and we agree with the reviewer about the remaining limitation regarding the dependency of overparameterization on the depth. Thank you again for your detailed review and for your recognition of our work's contribution to the field.

---

### Official Review · Reviewer_hUr4 · 2024-11-04

**Soundness:** 3
**Presentation:** 3
**Contribution:** 3
**Rating:** 8
**Confidence:** 2

**Summary:**

This paper obtains convergence and generalization bounds for gradient descent training of neural networks with smooth activation functions. An algorithmic stability analysis exploiting the Hessian properties in the gradient descent path allows to bound the generalization gap with the cumulative optimization error.  Ultimately, the bound on the test loss depends on the Euclidean distance with initialization and the Lipschitz constant of the network at initialization. Experiments with a 2-hidden layer network on MNIST and FashionMNIST show that the theoretical bounds provide accurate approximations for the test loss and generalization gap.  Under an NTK separability condition, the results of the paper (with smooth activations) lead to a better bound on the test loss then previous results with ReLU activation functions ($\sqrt{\frac{m}{n}}$ is replaced with $\frac{1}{n}$). Furthermore, on the specific problem of a d-dimensional XOR distribution, using quadratic activations with a large step size, only log(d) iterations (and constant width) are needed to obtain perfect test accuracy with n = O(d) samples. An experiment also confirms the logarithmic dependency on data dimension.

**Strengths:**

-The paper is clearly written and well organized.

-The paper provides novel convergence and generalization bounds for neural networks with smooth activations, which are significantly better than prior bounds derived for ReLU networks (the test loss bound obtained is width independent). This is important for understanding how generalization of neural networks with smooth activations compares to generalization of ReLU neural networks.

-The result showing efficient learning for the XOR problem with linear sample complexity is a notable contribution.

**Weaknesses:**

-Experiments on deep networks are done on a small neural network. Also, the expectations over S are estimated with one realization of S.

-The method to bound the spectral norm of the Hessian based on the distance from initialization is not new, limiting the novelty of the approach.

-The paper does not clarify whether the improved bounds are fundamentally due to the smoothness of the activation functions or if they result from limitations in the proof techniques used in previous bounds for ReLU networks, which may have led to weaker results in the non-smooth case.

**Questions:**

1) Do you think that the bounds holding for ReLu networks could be improved to match the bounds of your paper, or is the smoothness assumption necessary to obtain the improvement?

2) Could you clarify the main novel techniques or approaches used in the proofs of your main results, particularly in relation to previous work in this area?

---

> ### Author Response · Authors · 2024-11-19
>
> We thank the reviewer for their time and their valuable comments on our submission.
>
>
>  -We refer the reviewer to Remark 2.2 and lines 301-310 for a proof comparison of Thm. 2.1 with previous results. In summary, for the results in NTK regime, we used an induction-based argument to show that both $\|w_t-w_0\|$ and $\|w_t-w^*\|$ remain sufficiently small during training. This is essential to show that the bounds for the objective’s Hessian spectral-norm and the objective’s minimum eigenvalue remain valid.
>
>
> -The improvements mainly come from using an algorithmic-dependent analysis which bounds the generalization loss based on training performance whereas previous works used Rademacher-complexity bounds (Bartlett et al 2017,chen et al 2021) which captures the distance in L2,1 norm (please also see lines 128-168 in the Prior Works section). Whether the algorithmic-stability framework can be extended to neural nets with non-smooth activations is unknown and a possible direction for future studies. We believe such an extension is possible, although it requires new proof techniques as our current analysis relies on the smoothness assumption for both convergence and generalization.
>
>
> Q1- please see the last paragraph.
>
> Q2- In addition to our response in paragraph 1 above, the other problem that we study, learning the XOR, also goes beyond using known techniques. Let us  give a brief proof overview for Theorem 2.4 on the XOR problem. We first derive a generic expression for the expected weights at any iteration. We then decompose the error terms (due to SGD noise) in two components  where the first one  is the deviations of the empirical gradient around its expected value and the other is the deviations of the empirical gradient at true weights compared to expected weights. The calculations of error terms lead to the system of equations in section E.7.
> The most challenging part of the proof is in simplifying this system of equations as done in Section E.7. This needs careful treatment in order for the final noise strength (i.e., noise resulting from SGD) to be sufficiently small. For simplifying the error terms first we derive the $\ell_\infty $ norm of error terms and replace the resulting term in the third equation for obtaining the $\ell_2$ norm of the noise resulting from coordinates $k\in[3,d].$ To the best of our knowledge, the derivations and the approach above are novel.

---

> > ### Comment · Reviewer_hUr4 · 2024-12-03
> >
> > Thank you for the response. I will keep my score.

---

> > > ### Author Response · Authors · 2024-12-03
> > >
> > > Thank you again for your insightful and positive review and for highlighting the contributions of our work. If you have any further comments, please do not hesitate to let us know.

---

### Meta-Review · Area_Chair_TzxP · 2024-12-16

**Metareview:**

The work presents new advances on generalization bounds for learning neural networks with smooth activations. The approach is based on algorithmic stability, avoiding the challenges with Rademacher complexity based bounds, and also applicable in a small width regime, effectively extending the scope of NTK style analysis. The paper presents precise results under data separability with margin $\gamma$, with sharper results compared to past work. The paper also presents results on the noisy setting and also studies the special case of the XOR distribution. The reviewers overall appreciated the advances reported in the paper, including the new techniques and the implications of the results. Some concerns were brought up which have been mostly addressed by the authors during author discussion.

**Additional Comments On Reviewer Discussion:**

The reviewers have acknowledged the author responses and noted that their concerns have been mostly addressed during the discussion phase.

---

### Decision · Program_Chairs · 2025-01-22

Accept (Poster)